



# Fluxes from Soil Moisture Measurements (FluSM v1.0). A Data-driven Water Balance Framework for Permeable Pavements

Axel Schaffitel[1], Tobias Schuetz[2], Markus Weiler[1]

[1] Faculty of Environment and Natural Resources, University of Freiburg, Freiburg i. Br., 79098, Germany
[2] Department of Hydrology, University of Trier, Germany

*Correspondence to*: Axel Schaffitel (axel.schaffitel@posteo.de); Markus Weiler (markus.weiler@hydrology.uni-freiburg.de)

**Abstract.** Water fluxes at the soil-atmosphere interface are a key information for studying the terrestrial water cycle. However, measuring and modelling water fluxes in the vadose zone poses great challenges. While direct measurements require costly lysimeters, common soil hydrologic models rely on a correct parametrization, a correct representation of the involved processes and on the selection of correct initial and boundary conditions. In contrast to lysimeter measurements, soil moisture measurements are relatively cheap and easy to perform. Using such measurements, data-driven approaches offer the possibility to derive water fluxes directly. Here we present FluSM (Fluxes from Soil Moisture measurements), which is a simple, parsimonious and robust data-driven water balancing framework. FluSM requires only one single input parameter (the infiltration capacity) and is especially valuable for cases where the application of Richards based models is critical. Since Permeable Pavements (PPs) present such a case, we apply FluSM on a recently published soil moisture dataset to obtain the water balance of 15 different PPs over a period of two years. Consistent with findings from previous studies, our results show that vertical drainage dominates the water balance of PPs, while surface runoff plays only a minor role. An additional uncertainty analysis demonstrates the ability of the FluSM-approach for water balance studies, since input and parameter uncertainties have only small effects on the characteristics of the derived water balances. Due to the lack of data on the hydrologic behavior of PPs under field conditions, our results are of special interest for urban hydrology.



## 1.) Introduction

Soil moisture controls the partitioning of energy and water fluxes at the ground surface and is of major importance for understanding and modelling the terrestrial water cycle (Eagleson, 1978; Lahoz and De Lannoy, 2014; Trenberth and Asrar, 2014; Vereecken et al., 2015). Within the last decades, major technological progress has been made in terms of measuring soil moisture

at the point scale (see Robinson et al., 2008 for an overview over the various measuring techniques). Hence, long-term, high-resolution soil moisture data become increasingly available for studying soil hydrological processes (Vereecken et al., 2015). Such measurements have been used in different ways for modelling of water movement within the vadose zone, which is of major interest for understanding the terrestrial water cycle on different spatial scales.

Modelling of soil water fluxes is commonly achieved by using the Richards equation. However, the results of Richards based model approaches depend on the usage of valid soil hydraulic parameters, appropriate initial and boundary conditions and the selection of the right model dimensionality (Vereecken et al., 2010). Thereby, Richards based models may be inappropriate for fields, where it is difficult to determine representative soil hydraulic parameters and where the exact representation of different soil hydrological processes is unclear.

Simple bucket-type soil water balance approaches offer a promising alternative for modelling soil water fluxes. Since they are robust and parsimonious in terms of parameter demand (Vereecken et al., 2010), such approaches are common in land surface models and lumped catchment models (e.g. Albertson and Kiely, 2001; Boulet et al., 2000; Famiglietti and Wood, 1994; Rodriguez-Iturbe et al., 1999). In bucket-type approaches, soil is typically treated as a conceptual store (bucket) and fluxes into

and out of the bucket are calculated by governing equations (Vereecken et al., 2010). In literature, there are at least two different bucket-type approaches namely models and data-driven approaches. Thereby, models use governing equations to describe water fluxes into and out of the bucket and soil moisture measurements are often used to calibrate those models (e.g. Albertson and Kiely, 2001; Brocca et al., 2008). After calibration, bucket-type models enable to calculate soil moisture within the soil layer as well as incoming and outgoing water fluxes. However, one drawback is the dependency on the correct formulation of governing

equations. For applications where processes are unclear, data-driven approaches offer a promising alternative to study water fluxes since they do not rely on governing equations. These approaches use soil moisture and meteorological data as an input to infer the state of the soil storage. Based on conditional statements, the change of the soil storage over time is attributed to water fluxes. Such a data-driven approach was used for example by Breña Naranjo et al. (2011) to estimate evapotranspiration of a forest chronosequence. Weak points of their approach include the sensitivity of model results on the assumed bucket depth and the

conditional statement of vertical drainage ending 2 h after rainfall.

In the following, we present a new data-driven soil water balance framework which enables to derive water fluxes from soil moisture and meteorological measurements (hereinafter called FluSM (Fluxes from Soil Moisture measurements)). In contrast to



other data-driven approaches, FluSM derives the bucket depth directly from measurements and uses a simple and parsimonious concept for predicting vertical drainage explicitly. We think that FluSM is a valuable tool for cases where the application of Richards based modelling approaches is critical (e.g. due to limited parameter availability or unclear processes). Permeable pavements (PPs) are one example for such a case. Factors complicating the estimation of representative soil hydrological

parameters for PPs include the strong heterogeneity of urban soils and the coarse-grained soil material underneath the pavement layer. Furthermore, there are only limited data available for calibrating and validating soil hydrological models for PPs.

Studying the water balance of PPs is of major interest for urban hydrology, since PPs are expected to provide beneficial effects for urban hydrology (e.g. Andersen et al., 1999; Fassman and Blackbourn, 2010; Park et al., 2014; Scholz and Grabowiecki, 2007).

However, studies on the long-term hydrological behavior of PPs under realistic field conditions are sparse (see Timm et al. (2018) for a recent review). This is due to the high costs of water flux measurements (lysimeters), but also due to the difficulties of performing representative measurements within the urban environment. As a result, there is a knowledge gap concerning the hydrological processes involved at PPs. An example is the role of evaporation on PPs. While many authors assume that coarse grained soil layers limit evaporation of PPs (Brown and Borst, 2015; Fassman and Blackbourn, 2010; Flöter, 2006; van de Ven,

1990), other authors suppose that this is partially compensated by an increased water vapor flux due to increased vertical temperature gradients (Kodešová et al., 2014). In contrast to lysimeters, soil moisture measurements are cheap and easy to perform within urban areas (see Schaffitel et al., 2019 for an example). We therefore expect that such measurements, in conjunction with the FluSM-approach, lead an improved data basis for studying the water balance of PPs and further can be used for model calibration and validation purposes.

In this paper, we first explain the structure and the computational steps of FluSM and list its requirements. Afterwards, we introduce the individual calculation steps and benchmark the results obtained for 15 different PPs with literature values. Finally, we analyze the effects of uncertain inputs and parameters on the results of the FluSM-approach.





## 2.) Methods

### 2.1) FluSM structure and process representation

The flux and state variables considered within FluSM are depicted in Fig. 1 and explained in the following. At the ground surface, precipitation ($P$) first fills the surface storage, which has a defined capacity ($C_{surf}$). Water is removed from the surface storage

5    during subsequent dry periods by surface evaporation ($E_{surf}$). The fluxes $E_{surf}$ and $P$ determine the state of the surface storage ($S_{surf}$). Excess $P$ splits into surface runoff ($R$) and infiltration ($I$) depending on the infiltration rate ($I_{cap}$). Within the soil, the infiltrating water fills the soil storage which is emptied by the processes of evapotranspiration ($E$) and vertical drainage ($Q$). Hence, the state of the soil storage ($S_{soil}$) is controlled by the fluxes $I$, $E$ and $Q$.

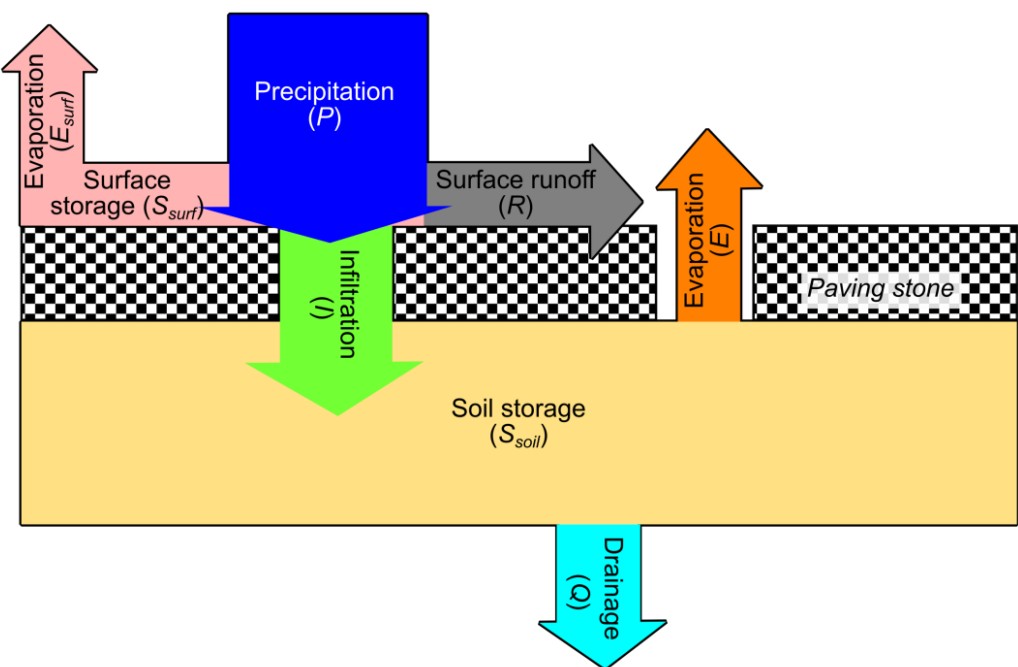

**Figure 1: Flux and state variables defined within FluSM.**





FluSM involves 4 different computational steps and requires time series of volumetric soil moisture ($\theta$), potential evaporation ($E_0$) and $P$ as inputs, as well as a plot specific $I_{cap}$ (Fig. 2). For the application of FluSM, there are some requirements concerning the input data and the soil hydrological behavior of the site. These requirements are briefly listed in the following:

- $\theta$ should be measured close to the ground surface
5   - $\theta$ time series should be corrected for fluctuations not related to water fluxes (e.g. due to temperature effects)
- Free vertical drainage

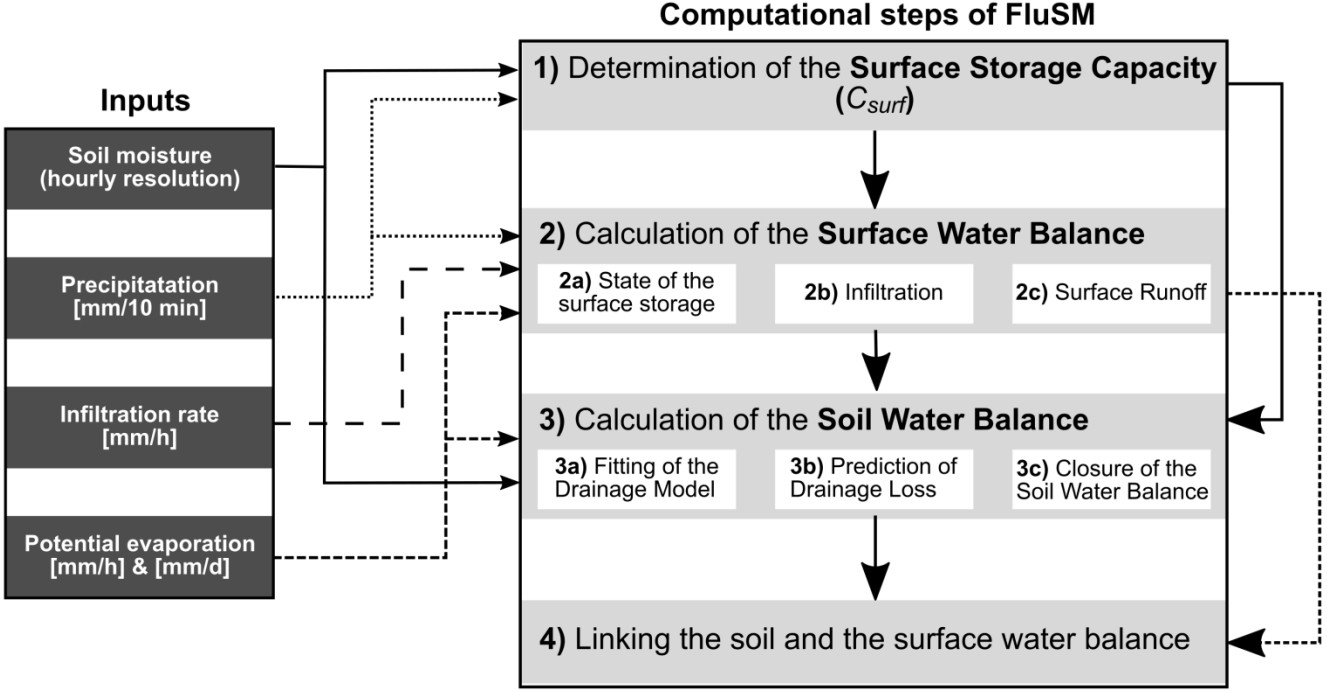

**Figure 2: Inputs and computational steps of FluSM.**

The computational steps work with an hourly resolution, except the surface water balance, which is calculated with a 10 min temporal resolution and aggregated to an hourly resolution at the end. In step 1 of FluSM, $C_{surf}$ is determined from time series of $P$ and $\theta$. Therefore, first the amplitude of the soil moisture response is calculated for each rain event. Afterwards, the events are grouped according to precipitation sums and the median soil moisture response is calculated for each precipitation class. For

15  classes below a specified response threshold, we assume that rainfall did not exceed $C_{surf}$. Based on this assumption, $C_{surf}$ is defined to be equal to the precipitation sum of the highest class not exceeding the response threshold (illustrated in Fig. 3). Consequently, $C_{surf}$ integrates the depth of water stored at the ground surface, but also the depth of water stored within the soil above the soil moisture sensor.





In step 2, the surface water balance is calculated as:

$$P(t) = \frac{dS_{surf}}{dt} + I(t) + R(t) \qquad \text{Eq. 1}$$

5   While $S_{surf}$ is below $C_{surf}$, rainfall only fills the surface storage (step 2a of FluSM). During subsequent dry periods, the surface storage is emptied by $E_{surf}$ with rates specified by the $E_0$. Thereby, we assume that $E_{surf}$ can be neglected during times of rainfall and further that there is no condensation of water vapor on the ground surface (times with negative $E_0$). In step 2b and 2c, the excess of $P$ divides into $I$ (Eq. 2) and $R$ (Eq. 3) and the partitioning between the fluxes is controlled by a plot specific $I_{cap}$, which is constant over time.

$$I(t) = \begin{cases} P(t) & if \ P(t) \leq I_{cap} \\ I_{cap} & if \ P(t) > I_{cap} \end{cases} \qquad \text{Eq. 2}$$

$$R(t) = \begin{cases} 0 & if \ P(t) \leq I_{cap} \\ P(t) - I_{cap} & if \ P(t) > I_{cap} \end{cases} \qquad \text{Eq. 3}$$

15   After calculating the surface water balance, the soil water balance is calculated in step 3 (Eq. 4).

$$\frac{dS_{soil}}{dt} = I(t) - E(t) - Q(t) \qquad \text{Eq. 4}$$

We use the measured change of the soil moisture ($d\theta/dt$) as a proxy for $dS_{soil}/dt$ (note that $d\theta/dt$ is negative for times with decreasing $\theta$, while it is positive for times with increasing $\theta$). By substituting $dS_{soil}/dt$ with $d\theta/dt$ in Eq. 4, soil water fluxes [mm/h]

20   are converted to soil moisture fluxes [vol.% /h]. Since measured $d\theta/dt$ integrates all soil moisture fluxes, the individual fluxes cannot be inferred directly from the measurements. We therefore describe $Q$ as a function of $\theta$ by using an unit gradient approach (Hillel, 1998) with the Burdine-Brooks-Corey parametrization of the unsaturated hydraulic conductivity (Brooks and Corey, 1964) (Eq. 5, hereinafter called drainage model).





$$Q = k_s * \left(\frac{\theta - \theta_r}{\theta_s - \theta_r}\right)^{\frac{2+3B}{B}} \qquad \text{Eq. 5}$$

Where $k_s$ is the saturated hydraulic conductivity, $\theta_r$ the residual water content, $\theta_s$ the saturated water content and $B$ the pore size distribution index. While $\theta$ is the only variable controlling $Q$, $E$ is also controlled by $E_0$. For dry periods with an atmospheric demand below a specified threshold, we assume evaporation to be negligible and thus $-d\theta/dt$ to be equal to $Q$. Therefore, we can use the soil moisture observations of these periods to fit the drainage model (Eq. 5) and to derive the parameters $k_s$ and $B$ and their

5     associated standard errors ($\sigma_{ks}$ and $\sigma_B$). In FluSM, the least-square optimization algorithm implemented in the Python module SciPy is used for the fitting procedure. Thereby, $\theta_r$ and $\theta_s$ are obtained from the minima and the maxima of the soil moisture time series.

In the next step, the calibrated drainage model is used to predict $Q$ from the measured soil moisture time series (FluSM step 3b). Assuming that $I$ and $E$ do not occur simultaneously, we obtain $I$ and $E$ by closing the mass balance (Eq. 6 and Eq. 7).

$$I(t) = Q(t) + \frac{d\theta}{dt} \qquad if \qquad -\frac{d\theta}{dt} < Q(t) \qquad \text{Eq. 6}$$

$$E(t) = -\frac{d\theta}{dt} - Q(t) \qquad if \qquad -\frac{d\theta}{dt} > Q(t) \qquad \text{Eq. 7}$$

We use two constraints to close the soil water balance, which are (1) no evaporation while water is available in the surface storage

15     and (2) no influx into the soil layer during dry periods (see Sect. 5 for an explanation and discussion of the constraints). For times where one constraint is active, $Q$ is assessed directly from the observed $-d\theta/dt$, while at the same time either $E$ (constraint 1) or $I$ (constraint 2) is set to 0.

Finally, the soil water balance and the surface water balance are linked by mapping infiltration calculated by the soil water balance on infiltration calculated by the surface water balance. In this step, a monthly variable scaling factor is determined from

20     the slope of a linear regression, which is performed between the cumulative infiltration obtained from the soil water balance and cumulative infiltration of the surface water balance. This scaling factors represents the depth of the soil bucket [mm], which is needed to infer $S_{soil}$ [mm] from $\theta$ [vol.%]. After determining the scaling factor, it is applied to transform all soil moisture fluxes into water fluxes.



## 2.2) Case study

For our case study, we used the data set provided by Schaffitel et al. (2019) which includes temperature corrected soil moisture times series, infiltrometer measurements and climate data, recorded within the city of Freiburg (Germany). At each PP included within the data set, soil moisture was measured in various depths over a 2-year lasting study period (Nov. 2016 – Oct. 2018). Using the soil moisture measurements, Schaffitel et al. (2019) classified the PPs into free-draining PPs and such with a restricted drainage behavior. For our case study, we only included the 15 free-draining PPs (hereinafter called plots), since this is a basic requirement for using FluSM. Schaffitel et al. (2019) further provides plot-specific values for $I_{cap}$ and initial and end infiltration rates ($I_{start}$ and $I_{end}$). Table 1 shows the properties and parameters of the 15 PPs included within this case study.





**Table 1: Characteristics of the PPs considered within the case study. For further information see** Schaffitel et al. (2019)**.**

| Name[1] | Plate15 | CP14 | CP13 | CP12 | CP11 | NP10 | NP9 | NP8 | NP7 | NP6 | NP5 | NP4 | NP3 | CP2 | GP1 |
|---|---|---|---|---|---|---|---|---|---|---|---|---|---|---|---|
| proportion of sealing | | | | | | | | | | | | | | | |
| Cluster | G | A | A | C | G | D | B | D | D | F | F | E | E | C | A |
| Sealing degree [%] | 97 | 93 | 92 | 89 | 86 | 86 | 83 | 81 | 80 | 78 | 76 | 76 | 75 | 73 | 54 |
| Build (year) | 1987 | 1999 | 1999 | 2004 | 1987 | 1998 | 2011 | 1998 | 1998 | 2014 | 2014 | 1999 | 1999 | 2004 | 1999 |
| Joint condition | 3 | 3 | 3 | 1 | 3 | 1 | 0 | 1 | 1 | 0 | 0 | 2 | 2 | 1 | 2 |
| $I_{cap}$ [mm/h] | 0.37 | 1.79 | 2.73 | 114.19 | 8.9 | 21.37 | 754.25 | 23.40 | 12.89 | 73.85 | 117.14 | 73.85 | 117.14 | 1133.29 | 109.92 |
| $I_{start}$ [mm/h] | 2.09 | 8.00 | 7.30 | 414.34 | 14.98 | 51.24 | 1483.52 | 23.40 | 25.19 | 158.8 | 381.51 | 158.8 | 381.51 | 2167.00 | 175.66 |
| $I_{end}$ [mm/h][2] | 0 | 0.60 | 1.80 | 80.40 | 7.20 | 15.00 | 591.00 | 23.40 | 10.20 | 55.20 | 58.20 | 55.20 | 58.20 | 903.00 | 95.40 |

1: The names include the type of PPs (Plate; CP: concrete paver; NP: natural paver and GP: grass paver) and a number, which is assigned according to the degree of surface sealing in a descending order

2: Assessed from the A parameter of the fitted Philip infiltration model (Schaffitel et al., 2019), which represents the theoretical minimum of the infiltration rate



For the city of Freiburg (Germany), data of four different climate stations are available (Schaffitel et al., 2019). For our case study (Sect. 3.1 and Sect. 3.2), we used the meteorological data of the WBI climate station, since this station is effected by the urban climate and data are free from gaps. Over the 2 year lasting study period, a total of 2440 mm of rainfall was recorded, while potential evaporation sums up to a total of 1570 mm. In order to estimate the uncertainty of the climate input data, we used data of all four climate stations (Sect. 3.3). Thereby, data of two climate stations exhibit data gaps, while the other two are free from data gaps. In order to obtain continuous time series for all four stations, data gaps were filled with the associated records of the WBI climate station. For our case study, the threshold for the soil moisture response (required for FluSM step 1) was set to 0.4 vol.%, which represents 4-times the resolution of the soil moisture sensors. Regarding the threshold for potential evaporation (required for FluSM step 2a) we used a value of 0.5 mm/day, which represents a compromise between a low evaporative demand and high number of observations for fitting the drainage model.



## 2.3.) Uncertainty analysis for the case study

Errors in hydrological modelling may arise from four different sources which are a) uncertain parameters, b) uncertain input variables, c) uncertainties in model structure and d) selection of data used for calibration (Deletic et al., 2012). For our case study, we developed an approach to quantify the effect of parameter and input uncertainty on the results of FluSM. In the

process, we first estimated uncertainties for each parameter and input variable and then propagated these through FluSM by means of 10,000 Monte-Carlo simulations generating the uncertainty range for each component of the water balance.

Uncertainties in the input variables ($\theta$, $P$ and $E_0$) may arise from systematic and from random errors. For the climatic input variables, only systematic errors were considered. Such errors may arise from a) the heterogeneity of the urban climatic

conditions and b) offsets in measurements. On the scale of an urban canyon, Koelbing et al. (2017) showed that the effect of a) on $E_0$ is mainly caused by the spatio-temporal variability of shading. Following Bitar (2004), we considered this uncertainty by a factor ranging between 0.5 and 1.4, which during the Monte-Carlo simulations was randomly resampled from a uniform distribution. Due to the large range considered, the analysis should also account for uncertainties in estimating $E_0$. For rainfall, a certain degree of the spatial heterogeneity is captured by the variability among the four climate

stations within the study area. We accounted for this variability by using the four $P$-time series as ensembles for the Monte-Carlo simulations. Additionally, we multiplied the rainfall time series by a factor ranging between 0.8 and 1.2 (uniform distribution) to account for measurement bias and small-scale rainfall variability (e.g. within an urban canyon). Regarding the soil moisture measurements, systematic errors are irrelevant since all calculations are based on the relative change of $\theta$ over time rather than on absolute $\theta$. Due to the temperature correction schemes applied by Schaffitel et al. (2019) we expect

that random errors were removed to a great extent. Therefore, errors of soil moisture were not considered within the uncertainty analysis.

Uncertainties also exist for the parameter $I_{cap}$ and the two parameters of the drainage model ($k_s$ and $B$). For $I_{cap}$, we specified these uncertainties to range between the plot-specific $I_{start}$ and $I_{end}$ (Table 1). Parameter uncertainty of $k_s$ and $B$ was assessed

by their standard errors ($\sigma_{ks}$ and $\sigma_B$) obtained during the fitting procedure. Within the Monte-Carlo framework, we used uniform random resampling for all parameters.





# 3.) Results

## 3.1) Processes in the FluSM approach

First, we derived the plot specific $C_{surf}$. For our data set, the uppermost sensor was located directly beneath the paving layer. Hence, $C_{surf}$ comprises the amount of water stored at ground surface (depression storage and wetting capacity) and the amount of water stored within the joint material. Fig. 3 shows the classes of median soil moisture responses and reveals a $C_{surf}$ of 2.5 mm for plot CP14 (see Appendix A for the $C_{surf}$ of the other plots of the case study). The filling and emptying of the surface storage over 3 wet-dry cycles is depicted in Fig. 4.

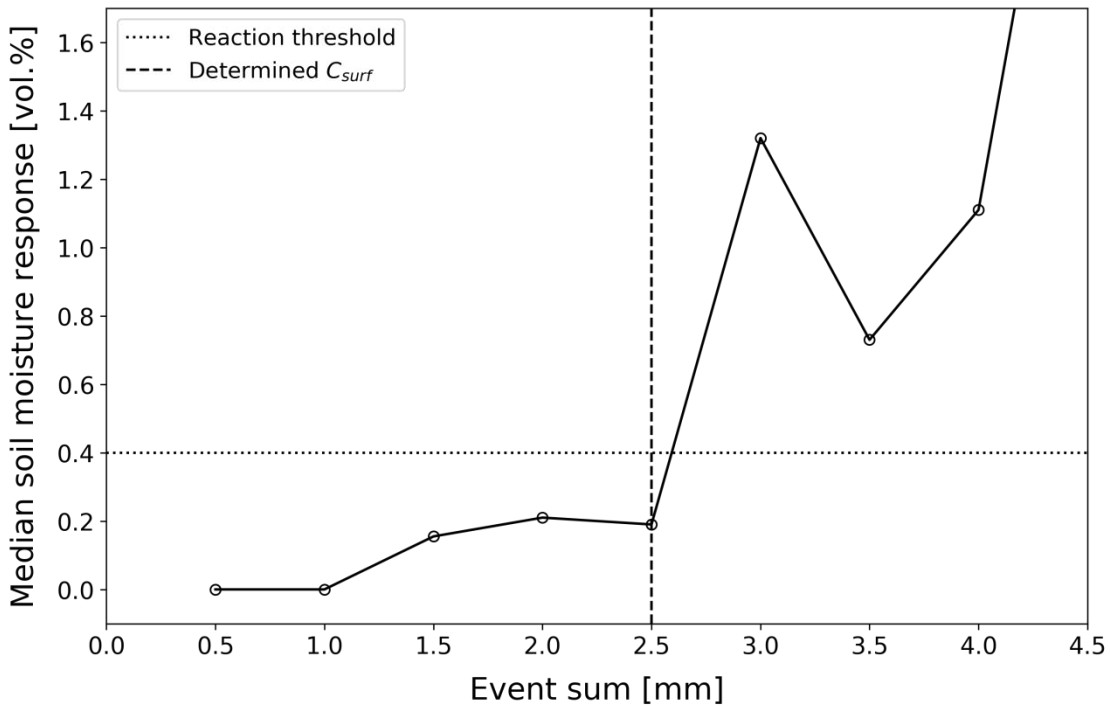

**Figure 3: Determination of $C_{surf}$ for plot CP14 (FluSM step 1).**





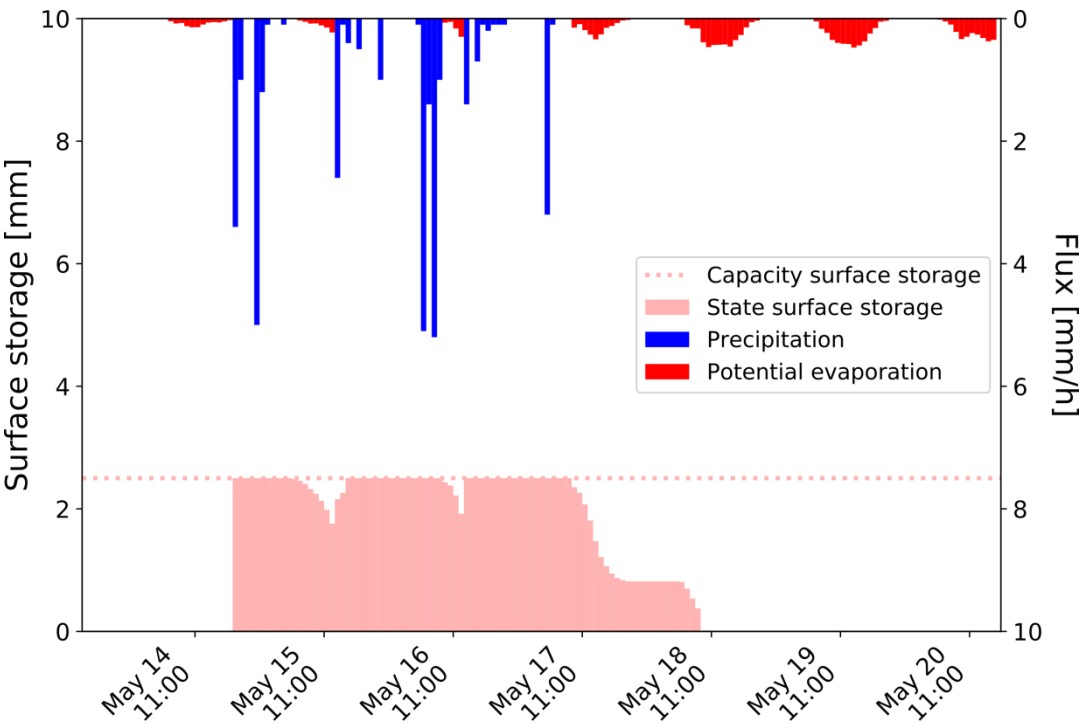

**Figure 4: Filling and emptying of the surface storage over 3 separated wet-dry cycles in May 2018 at plot CP14 (FluSM step 2a).**

During each event depicted in Fig. 4, the surface storage was entirely filled, while it was at least partially emptied during
5   subsequent dry periods. When the surface storage is entirely filled, the parameter $I_{cap}$ determines the partitioning of excess $P$
into infiltration and surface runoff. For plot CP14, $I_{cap}$ is quite low with 1.79 mm/h, but rainfall intensities measured at the
WBI climate exceed 2 mm/h only for around 27% the times with observed rainfall. Although, the formation of surface runoff
is rare, its quantity is still considerable at this plot. Fig. 5 shows the derived surface water balance for plot CP14 with around
40% of precipitation input leading to surface runoff (approx. 720 mm over the study period). Infiltration still dominates the
10   surface water balance with around 830 mm. Surface evaporation accounts for around 290 mm (16% of the precipitation
input).



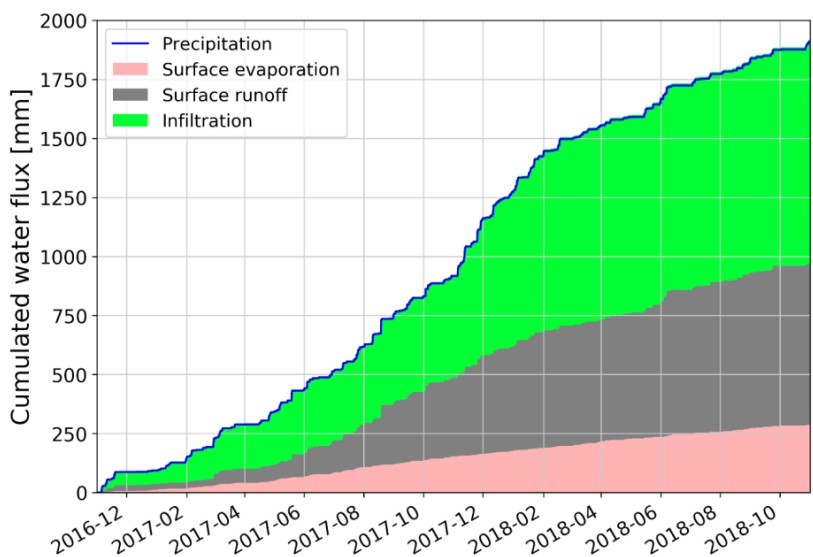

**Figure 5: Cumulative surface water balance of plot CP14 over the entire study period.**

For the soil water balance, fist the parameters of the drainage model (Eq. 5) were determined by fitting of the drainage model

5    to soil moisture observations made during periods with a low potential evaporation (Fig. 6).

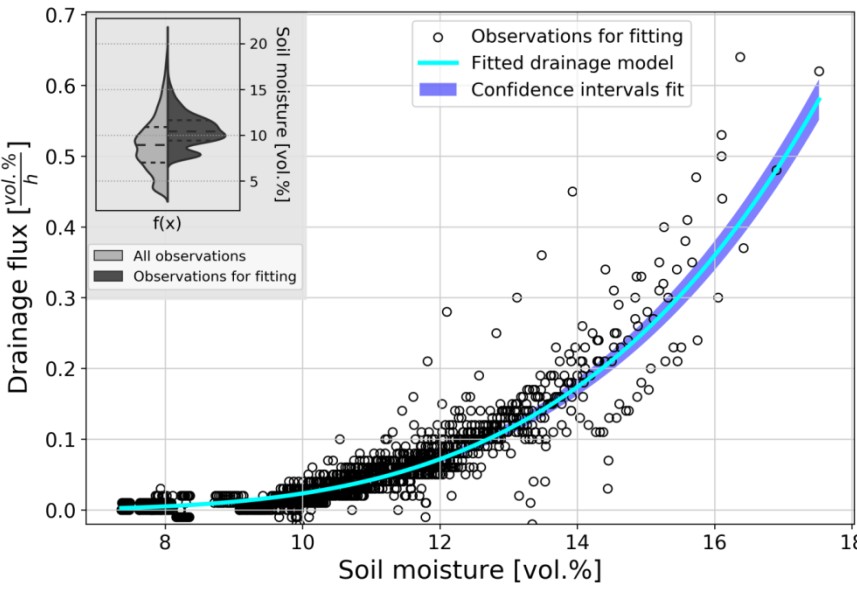

**Figure 6: Fitting of the drainage model for plot CP14. The inset shows the frequency distributions of all soil moisture observations and the observations used for fitting of the drainage model.**



The parameters obtained for plot CP14 account 1.44 ± 0.05 vol.% for $k_s$ and 1.78 ± 0.1 for B. The root mean squared error of the fit is 0.04 vol.%/h, which indicates a good quality of the fit. This is also reflected in the narrow confidence intervals depicted in Fig. 6. Additionally, the inset of Fig. 6 shows the density of the observations used for fitting (right side) and the entire population of all observations (left side). Although the distributions differ, the observations used for fitting cover the

5     range of the entire population to a large extent. Results for the other plots are reported in Table 4 of Appendix A.

In the next step of applying the soil water balance, the fitted drainage model was used to predict drainage rates from the measured soil moisture time series (FluSM step 3b). By using predicted drainage rates and the observed change of soil moisture, the soil water balance was closed. This is exemplarily shown in Fig. 7 for a period of 30 h for plot CP14. Thereby,

10    the effect of the model constraints become visible as surface evaporation starts only after the surface storage is completely empty and no water is infiltrating after the end of the rainfall event. The entire soil water balance of plot CP14 is shown in Fig. 8. For this plot, evaporation accounts for around 10% of the infiltrating water, while drainage dominates with 90%.

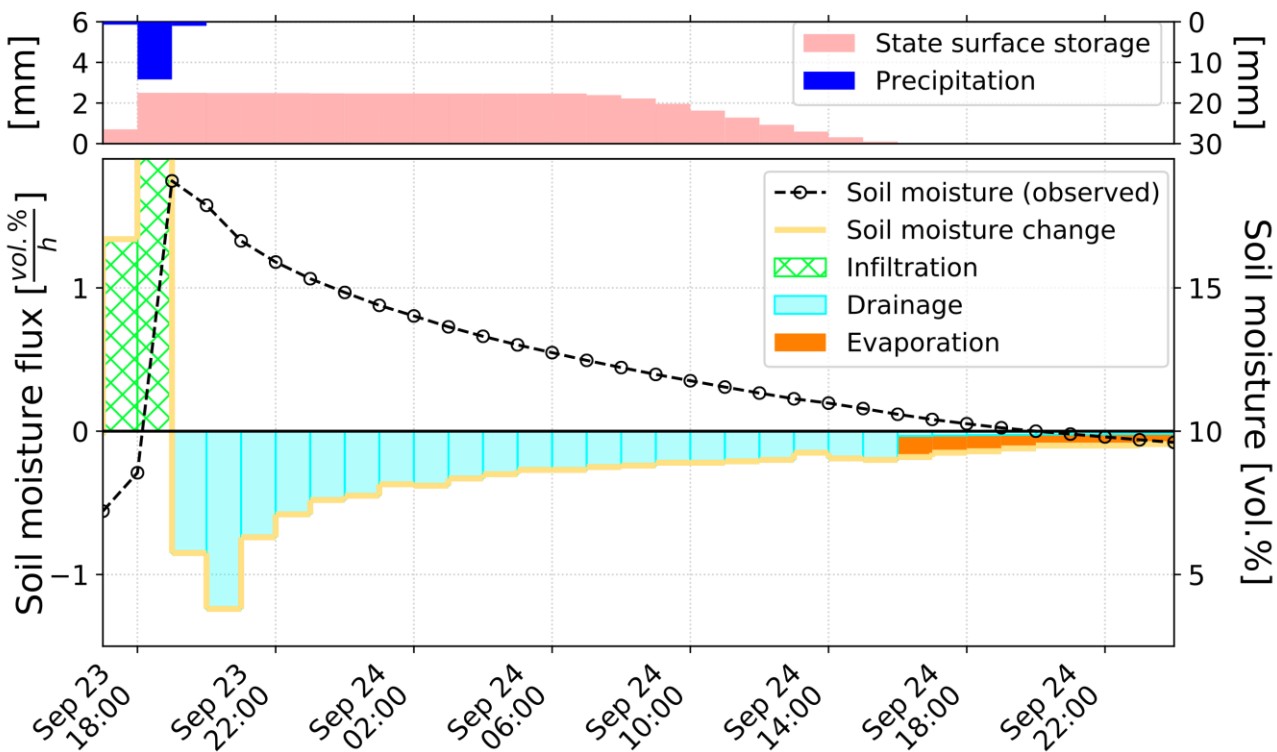

**Figure 7: Closure of the soil water balance from measured $d\theta/dt$ and predicted drainage loss for plot CP14 over a period of 30 h.**





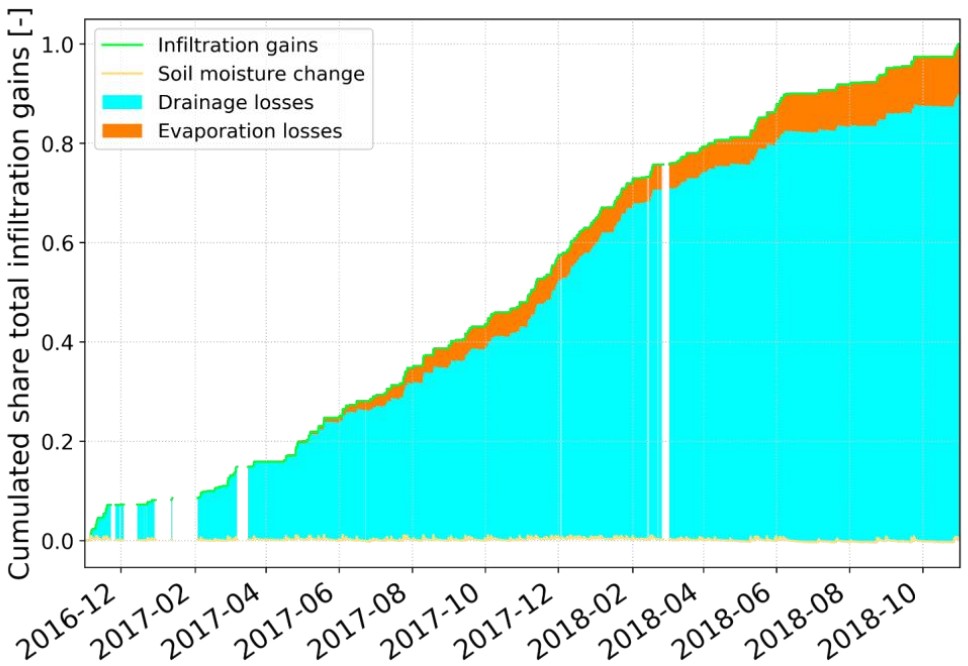

**Figure 8: Soil water balance for plot CP14. Gaps in the soil water balance are due to gaps in the recorded soil moisture time series.**

In FluSM step 4, the depth of the soil bucket [mm] is derived on a monthly time basis. For plot CP14, the median of the
5   monthly variable bucket depths accounts 60 mm with a standard deviation of 28 mm. Values for the other plots of the case
study are provided in Appendix A. Applying the bucket depths on the soil water balance, the entire water balance was
obtained (Fig. 9). The transformation of soil moisture fluxes into water fluxes leads to minor deviations between the total
precipitation input and the summarized water fluxes. For plot CP14, these deviations are negligible since they account for
less than 0.3% of the total water balance. While the surface water balance of plot CP14 revealed a total infiltration of
10   830 mm (Fig. 5), the complete water balance shows that most of this infiltration leads to drainage, which forms the largest
part of the water balance (770 mm). Furthermore, the water balance shows a total evaporation of 350 mm which is approx.
19% of the precipitation input.



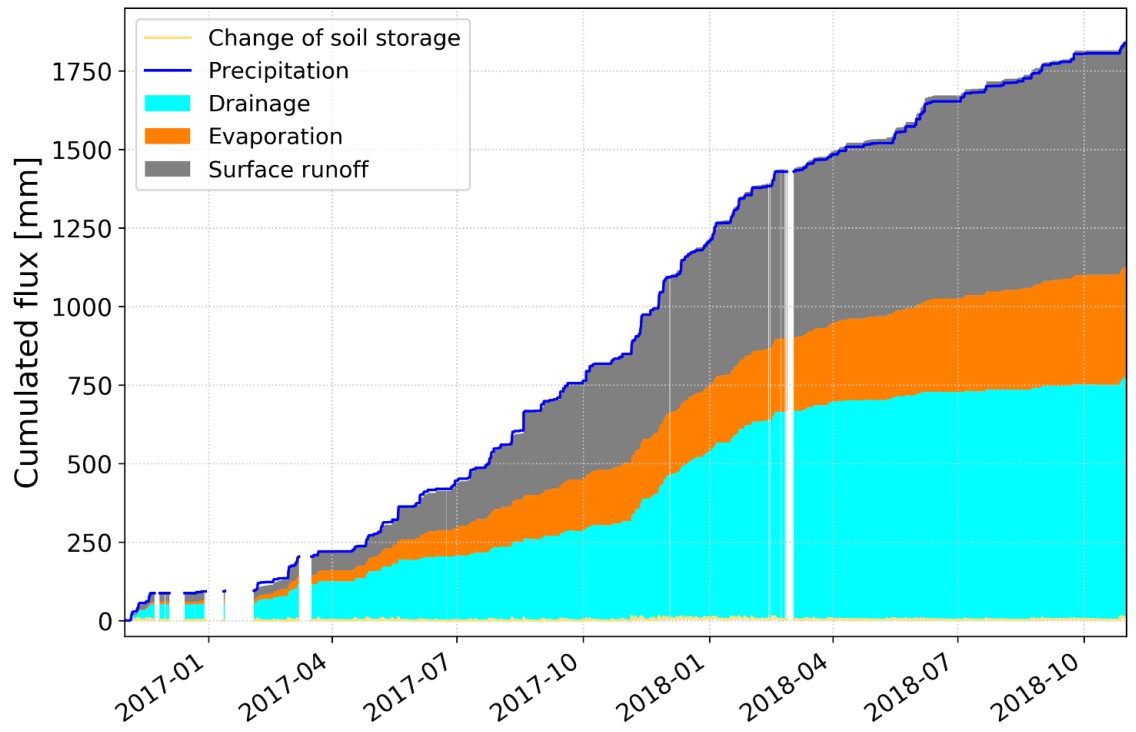

**Figure 9: Cumulated water fluxes of the entire water balance for plot CP14**





### 3.2) Benchmarking of FluSM for the case study

Validating the fluxes calculated for the PPs of the case study is challenging, since we did not measure the fluxes directly. To overcome this problem, we conducted a literature review on long-term water fluxes reported for PPs and compared them with our results. Since the water balance of PPs is controlled by various factors and since these factors vary between the

different studies, this comparison can serve only as rough validation. Foremost amongst the factors which control the water balance of PPs are local microclimatic conditions, soil properties, water availability and surface properties.

Studies on the long-term water balance of PPs under realistic field conditions are scarce. Known studies include lysimeteter measurements (Flöter, 2006; Rim, 2011; Timm, 2019; Wessolek and Facklam, 1997), drainage measurements (Brown and

Borst, 2015) and combined soil moisture and drainage measurements (Ragab et al., 2003). A recent review by Timm et al. (2018) further includes results of non-published studies. However, none of these studies includes more than 6 different PPs and plot specific parameters (e.g. degree of surface sealing, infiltration rate, surface clogging) are poorly documented. Given the differences in the climatic conditions among the studies, it is hardly feasible to make a general statement on the long-term water balance of PPs within the urban environment. In addition, Illgen (2009) showed that infiltration rates measured at

same type PPs can range over at least one order of magnitude. Table 2 shows the results found in literature on the long-term water balance of PPs.

**Table 2: Values reported in literature for the long-term water balance of PPs**

| | Type | Sealing degree [%] | Coefficient | | | Rainfall [mm] | Potential evaporation [mm] | measuring period | Location | Experimental setup |
|---|---|---|---|---|---|---|---|---|---|---|
| | | | Surface runoff [-] | Drainage [-] | Evaporation [-] | | | | | |
| **Flöter (2006)** | CP | 95 | 0.12 | 0.8 | 0.08 | n.a. | n.a. | 1996-2005 | Hamburg (Germany) | Lysimeter (non-weighable) |
| | Plate | 98 | 0.41 | 0.54 | 0.05 | n.a. | n.a. | 1996-2005 | Hamburg (Germany) | Lysimeter (non-weighable) |
| **Rim (2011)** | NP | 60 | 0.16 | 0.71 | 0.16 | 537 | n.a. | 03/2009-04/2010 | Berlin (Germany) | Lysimeter (weighable) |
| | Plate | 93 | 0.27 | 0.65 | 0.1 | 537 | n.a. | 03/2009-04/2010 | Berlin (Germany) | Lysimeter (weighable) |
| **Wessolek and Facklam (1997)** | GP | 59 | 0.05 | 0.49 | 0.46 | 631 | 605 | 04/1985-03/1986 | Berlin (Germany) | Lysimeter (non-weighable) |
| | CP | 97 | 0.17 | 0.61 | 0.21 | 631 | 605 | 04/1985-03/1986 | Berlin (Germany) | Lysimeter (non-weighable) |
| | Plate | 97 | 0.32 | 0.35 | 0.13 | 631 | 605 | 04/1985-03/1986 | Berlin (Germany) | Lysimeter (non-weighable) |
| **Timm & Wessolek** (Timm, 2019) | NP | 60 | 0.03 | 0.65 | 0.36 | 389 | 732 | 06/2016-06/2017 | Berlin (Germany) | Lysimeter (weighable) |
| | Plate | 93 | 0.16 | 0.64 | 0.23 | 405 | 732 | 06/2016-06/2017 | Berlin (Germany) | Lysimeter (weighable) |
| **Diestel & Schmidt (as cited in Timm et al., 2018)** | NP | n.a. | 0.07 | 0.74 | 0.19 | 575 | n.a. | n.a. | n.a. | Lysimeter (non-weighable) |
| | NP | n.a. | 0.09 | 0.67 | 0.24 | 575 | n.a. | n.a. | n.a. | Lysimeter (non-weighable) |
| | CP | n.a. | 0.1 | 0.78 | 0.12 | 575 | n.a. | n.a. | n.a. | Lysimeter (non-weighable) |
| | GP | n.a. | 0.06 | 0.68 | 0.26 | 575 | n.a | n.a. | n.a. | Lysimeter non-weighable) |





Table 3 shows the summarized water fluxes for all plots in our case study. Note that times with data gaps in soil moisture data were excluded from FluSM calculations resulting in lower rainfall amounts. In agreement with all available other studies (Table 2), the results of our case study show that the drainage flux dominates the water balance for all PPs except for plot CP15. For PPs with an $I_{cap}$ above 9 mm/h, surface runoff coefficients were below of 10%, while PPs with $I_{cap}$-values above approx. 70 mm/h showed negligible surface runoff (below 12 mm). Only at plots with $I_{cap}$ below 3 mm/h (CP15, CP14 and CP13) surface runoff is a major component of the water balance accounting for 30-80% of precipitation.

**Table 3: Summarized fluxes over the entire measuring period for all plots (evaporation includes $E$ and $E_{surf}$).**

| | | CP15 | CP14 | CP13 | CP12 | CP11 | NP10 | NP9 | NP8 | NP7 | NP6 | NP5 | NP4 | NP3 | CP2 | GP1 |
|---|---|---|---|---|---|---|---|---|---|---|---|---|---|---|---|---|
| **Rainfall [mm]** | | 1850 | 1838 | 1840 | 1153 | 1850 | 1792 | 1753 | 1792 | 1792 | 1796 | 1851 | 1796 | 1851 | 1153 | 1836 |
| **Surface runoff** | total [mm] | 1447 | 718 | 538 | 0 | 199 | 80 | 0 | 70 | 141 | 11 | 3 | 11 | 3 | 0 | 4 |
| | coefficient [-] | 0.78 | 0.39 | 0.29 | 0.00 | 0.11 | 0.04 | 0.00 | 0.04 | 0.08 | 0.01 | 0.00 | 0.01 | 0.00 | 0.00 | 0.00 |
| **Drainage** | total [mm] | 248 | 772 | 942 | 916 | 1185 | 1448 | 1305 | 1524 | 1337 | 1493 | 1525 | 1479 | 1494 | 802 | 1184 |
| | coefficient [-] | 0.13 | 0.42 | 0.51 | 0.79 | 0.63 | 0.79 | 0.73 | 0.84 | 0.73 | 0.80 | 0.80 | 0.80 | 0.78 | 0.69 | 0.62 |
| **Evaporation** | total [mm] | 186 | 348 | 345 | 188 | 421 | 246 | 469 | 166 | 258 | 283 | 309 | 305 | 339 | 313 | 600 |
| | coefficient [-] | 0.10 | 0.19 | 0.19 | 0.19 | 0.23 | 0.16 | 0.28 | 0.10 | 0.16 | 0.19 | 0.19 | 0.19 | 0.21 | 0.28 | 0.34 |

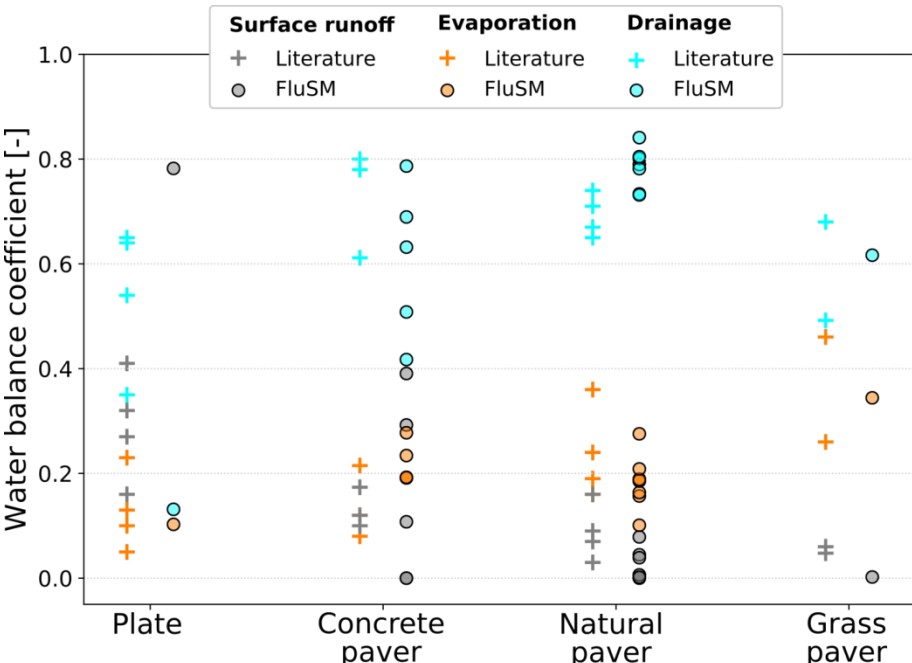

**Figure 10: Range of long-term water fluxes reported in literature for different PP types compared with the results of our case study.**





In Fig. 10, our results are compared to the long-term water balance reported in the literature for different types of PPs. For plates, the 4 studies found in literature show a lower runoff and a higher drainage coefficient compared to the plate included in our case study. This is due to the low infiltration rate measured at our plate (0.37 mm/h), which lies around 10 times

5 below the rates measured by Wessolek and Facklam (1997) for their plate lysimeter. For concrete paving stones, patterns of our results are comparable with literature values, although our results show a larger variation for this group. This is mainly due to the large range of infiltration rates which we measured for this group (ranging from 1.79 mm/h to 1133 mm/h). Nevertheless, the results consistently show that drainage dominates the water balance for all concrete pavers. Also for natural pavers, our results and literature values show equal patterns with drainage dominating the water balance, followed by

10 evaporation, while surface runoff accounts for the smallest part of the water balance. Compared to literature values, we predicted slightly higher drainage and at the same time smaller runoff and evaporation. For grass pavers our results for drainage and evaporation lie within the range of the literature values.

In general, our results are consistent with the patterns found by previous studies. Due to the small number of studies on long-

15 term water fluxes of PPs, but also due to the different environmental conditions, a direct validation of the FluSM results is difficult. Such a direct validation would only be possible for sites where water fluxes and $\theta$ were measured simultaneously.





### 3.3) Uncertainty analysis

We performed 10,000 Monte-Carlo simulations with FluSM in order to quantify the effect of uncertain inputs and parameters on the results of the case study. Based on our assumption of input uncertainty, precipitation ranged between 550 and 1150 mm/year, while potential evaporation ranged between 400 and 1150 mm/year (Fig. 11b). Hence, we considered a

5    wide range of climatic inputs, which should cover the variability of urban microclimatic conditions and further account for measurement errors. From the Monte-Carlo simulations, we obtained distributions for each water balance coefficient (Fig. 11a). In the following, we define uncertainty ranges as the distance between the 5- and the 95-percentile of the distributions.

**Figure 11:** Violin plots of water balance coefficients obtained from the 10,000 Monte-Carlo simulations (a) and range of the
10    considered variability of the climatic input (b). Points indicate the median of all runs, while horizontal lines represent the 5 and 95 percentiles of the distributions.





Even when considering uncertain inputs and parameters, the aforementioned patterns of drainage forming the largest water balance component at almost all PPs, evaporation accounting for at least 10% of the water balance and surface runoff being negligible for PPs with an average infiltration rate above 70 mm/h remain unchanged. Uncertainty ranges for evaporation coefficients account for ±6% of the total water balance and are similar for all PPs. Since predicted uncertainties for

5 evaporation are small, the considered input and parameter uncertainties have only a small effect on this water balance component. In contrast, uncertainties in surface runoff and drainage coefficients differ among the individual plots. These differences can be attributed to the parameter uncertainty of the infiltration rate, which we defined to range between the plot-specific initial and end infiltration rate. While the results are very robust for plots with high infiltration rates (e.g. NP4, NP3, CP2 and GP1), there are large uncertainties for PPs with low infiltration rates (e.g. CP15, CP14 and CP13). For PPs with an

10 infiltration rate above 70 mm/h, the uncertainty of this parameter has no effect on the water balance, since surface runoff is negligible even when end infiltration rates are used for FluSM.





## 4.) Discussion

### 4.1) Requirements and applicability of the FluSM-approach

Basic requirements for FluSM were listed in Sect. 2. In the following, we discuss these requirements and the conditions under which FluSM should produce reliable results. Due to the parsimonious approach of a constant infiltration rate, FluSM
is applicable for locations where it is difficult to estimate suction heads or matrix potentials. Our uncertainty analysis showed that the infiltration rate is a source of substantial errors for plots with a low infiltration rate, while it does not affect plots with a high infiltration rate. Therefore, the FluSM-approach is best suited for plots with a high infiltration rate. Since national regulations for PPs require an infiltration rate above 97.2 mm/h (Borgwardt, 2001), the FluSM approach is well suited for PPs which meet this requirement. Within FluSM, the parameter surface storage capacity is derived from soil moisture
observations and represented by a constant value. However, for sites covered by trees, this concept might be problematic since the surface storage capacity may be affected by a seasonally variable canopy coverage (Link et al., 2004). Under such conditions, we recommend to use rainfall data which represents the input at the ground surface (throughfall). As an alternative, the FluSM algorithm might be adapted to derive seasonal variable surface storage capacities.

Various bucket-type water balance models use a unit-gradient approach to describe drainage out of the soil bucket (e.g. Albertson and Kiely, 2001; Famiglietti and Wood, 1994; Rodriguez-Iturbe et al., 1999). An intrinsic assumption of the unit-gradient approach is that drainage occurs freely (no impeding layers) with gravity-driven rates (minor influence of matrix potential). Therefore, FluSM should only be used only for sites that meet these assumptions. Since the parameters of the drainage model are derived by fitting, the soil water balance relies on the quality of the fit. In order to obtain reliable fits, the
number of observations should be high. Therefore, soil moisture measurements should cover at least one entire year including at least one season with low atmospheric demand.

For closing the soil water balance, we used two different constraints. The first constraint limits evaporation to occur only from one layer at a time (no soil evaporation while there is water available in surface storage). This constraint is reasonable
for bare soils and sites with low growing vegetation. In contrast, for sites with tall vegetation, evaporation may occur simultaneously from the soil and from the interception storage. In order to apply FluSM for such sites, this constraint should be adapted. The second constraint limits fluxes into the soil layer to rain events. Since this constraint is only reasonable for layers located close to the ground surface, soil moisture data should originate from shallow depths. A further reason for using shallow soil moisture data is that evaporation acts mainly on the upper soil layers. Prior to the usage, we recommend to
preprocess soil moisture time series in order to correct for fluctuations not related to water fluxes. Such fluctuations may originate from the temperature-dependency of soil moisture measurements (Kapilaratne and Lu, 2017; Qu et al., 2013; Wraith and Or, 1999). An efficient correction method was presented by Schaffitel et al. (2019).





The results of bucket-type water balance models were shown to be very sensitive on the assumed soil (bucket) depth (Boulet et al., 2000; Breña Naranjo et al., 2011). Since FluSM derives this parameter directly from measurements, it does not rely on an assumption, which is a major advantage. Furthermore, the bucket depth is variable on a monthly time scale which enables

to account for processes which are variable over time (e.g. rooting depth, temperature dependency of water fluxes, evaporation depth).

## 4.2) Results for the case study

Previous studies showed the relevancy of groundwater recharge for PPs (Flöter, 2006; Rim, 2011; Scholz and Grabowiecki,

2007; Wessolek and Facklam, 1997). Our case study revealed a consistent picture as vertical drainage forms a major part of the water balance. This is remarkable, since urbanization is often assumed to reduce direct groundwater recharge (e.g. Brattebo and Booth, 2003; Göbel et al., 2004; Shuster et al., 2005). Nevertheless, the increased recharge on PPs may endanger the groundwater resource since PPs may facilitate the transport of pollutants to the subsurface (Scholz and Grabowiecki, 2007). Furthermore, an increased groundwater recharge might also be problematic for areas with a high

groundwater table (Göbel et al., 2004; van de Ven, 1990).

One possibility to reduce the groundwater recharge on PPs is to increase evaporation. The results of the case study showed that this component only accounts for approx. 10-35% of the water balance. The grass paver showed highest evaporation (approx. 35% of the water balance) of all PPs in this case study. Thereby, transpiration from vegetated voids, but also the

low degree of surface sealing should account for the elevated evapo(transpi)ration at this plot. For surface runoff, our results showed that this component is negligible for most PPs. Only for PPs with a very low infiltration rate, surface runoff accounted for more than 10% of the water balance. Since national regulations for PPs require a high infiltration rate (Borgwardt, 2001), PPs can be regarded as an efficient tool for mitigating urban runoff volumes. Nevertheless, various authors showed that clogging of joints causes a rapid decline of infiltration capacities over time (e.g. Boogaard et al., 2014b;

Lucke and Beecham, 2011). As a result, infiltration capacities of PPs can range over at least one order of magnitude (Illgen, 2011). In order to maintain high infiltration capacities on PPs, maintenance measures (e.g. street sweeping and vacuum cleaning) should be carried out (Boogaard et al., 2014a; Scholz and Grabowiecki, 2007; Winston et al., 2016).



## 5.) Conclusion

We presented the data-driven water balance framework FluSM, which allows to derive water fluxes directly from soil moisture and meteorological measurements. FluSM is parsimonious, as it relies only on one single input parameter - the site-specific infiltration rate. In contrast to other data-driven approaches, FluSM derives the soil (bucket) depth internally from

the input data and therefore the results do not depend on an assumed bucket depth. Furthermore, drainage is represented explicitly by a simple concept, which is more realistic than the derivation by a conditional statement. Due to its parsimony, FluSM is especially applicable for locations with limited soil hydrologic knowledge and limited data availability. Since permeable pavements are such locations, we used FluSM to derive their water balance. As a result, we obtained water fluxes of 15 different permeable pavements over a period of 2 years. So far, high resolution, long-term water fluxes of permeable

pavements were only obtained from lysimeter measurements. In contrast, the FluSM approach presents an easy and cheap alternative, since it requires only soil moisture and meteorological data. For PPs, FluSM proved its plausibility since the derived water balances are in line with findings from previous lysimeter studies. In addition, FluSM is also applicable for locations which are not suited for lysimeters.

**Appendix A**

**Table 4: Parameters obtained for the plots of the case study. For drainage model, the standard deviation of the parameters and the root mean squared error (RMSE) of the fit is included. For the monthly scaling factors, the median and the standard deviation is reported.**

|  | CP15 | CP14 | CP13 | CP12 | CP11 | NP10 | NP9 | NP8 | NP7 | NP6 | NP5 | NP4 | NP3 | CP2 | GP1 |
|---|---|---|---|---|---|---|---|---|---|---|---|---|---|---|---|
| $C_{surf}$ [mm] | 1.00 | 2.50 | 2.50 | 1.00 | 1.50 | 1.00 | 3.00 | 0.50 | 1.50 | 2.00 | 2.00 | 2.00 | 2.00 | 4.50 | 4.00 |
| $k_s$ [vol.%/h] | 20.09 ± 0.91 | 1.44 ± 0.05 | 1.7 5± 0.19 | 11.57 ± 0.89 | 5.43 ± 0.28 | 5.04 ± 0.23 | 23.29 ± 1.62 | 77.65 ± 5.29 | 8.27 ± 1.27 | 2.17 ± 0.04 | 4.41 ± 0.15 | 3.99 ± 0.24 | 4.03 ± 0.21 | 5.11 ± 0.58 | 0.32 ± 0.35 |
| $B$ [-] | 0.61 ± 0.01 | 1.78 ± 0.10 | 1.16 ± 0.11 | 0.68 ± 0.03 | 0.53 ± 0.01 | 1.23 ± 0.06 | 0.60 ± 0.02 | 0.52 ± 0.01 | 2.64 ± 0.52 | $3*10^7$ ± $8*10^{12}$ | 28.68 ± 14.61 | 2.62 ± 0.23 | 2.56 ± 0.21 | 0.56 ± 0.03 | 0.68 ± 0.04 |
| RMSE [vol.%/h] | 0.07 | 0.04 | 0.05 | 0.06 | 0.05 | 0.05 | 0.03 | 0.12 | 0.08 | 0.03 | 0.03 | 0.03 | 0.04 | 0.05 | 0.04 |
| Bucket depth (median) [mm] | 13.19 | 60.54 | 91.88 | 90.86 | 87.78 | 78.53 | 113.02 | 33.99 | 74.78 | 101.45 | 95.84 | 128.41 | 123.19 | 96.17 | 185.77 |
| Bucket depth (standard deviation) [mm] | 0.93 | 7.88 | 18.27 | 20.68 | 29.64 | 10.43 | 48.60 | 1.60 | 17.47 | 6.92 | 8.10 | 21.78 | 13.81 | 120.96 | 482.46 |

**Code availability**

FluSM was written in Python 2.7.15 and is provided through a Gitlab repository (https://gitlab.com/ASchaffitel/flusm). The code is open source and provided under the terms of the GNU General Public License v3. FluSM is structured into 4 different modules which are called by the FluSM.py module. Detailed information on structure and usage of FluSM is



provided by a readme file on the Gitlab repository. Furthermore, the Gitlab repository also includes the sample data used for the case study.

**Acknowledgements**

5  We would like to thank Anne Timm and Gerd Wessolek from the Technical University Berlin (Germany), Department of Ecology, Soil Conservation for providing lysimeter data.

**Author Contributions**

AS, TS and MW conceived the idea behind FluSM. AS developed FluSM and prepared the manuscript with contributions from all co-authors.

**Financial support**

This study is part of the research project WaSiG (Wasserhaushalt siedlungsgeprägter Gewässer), which is part of the joint project ReWaM (Regionales Wasserressourcen-Management für den nachhaltigen Gewässerschutz in Deutschland) funded by the German Ministry of Education and Research (BMBF), grant no. 033W040B. Furthermore, this work was supported

15  by the badenova AG and Co. KG ("Innovation Fund for the Protection of Climate and Water"), project number 2015-08.



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
