# Peer review of "Fluxes from Soil Moisture Measurements (FluSM v1.0). A Data-driven Water Balance Framework for Permeable Pavements"

_Geoscientific Model Development, 2020_

## Referee Comment (RC1) · Anonymous Referee #1 · 20 Jul 2020

**OVERVIEW**

The study describes the development of a data-driven approach for estimating water fluxes from soil moisture observations. The method is tested over 15 different permeable pavements in the city of Freiburg.

**GENERAL COMMENTS**

The paper is mostly well written and clear. The topic of the paper is interesting for the readership of GMD as a simplified approach for estimating water fluxes from soil moisture observations is a novel approach and potentially important approach as being cost effective. Moreover, the spread of soil moisture sensors worldwide potentially

guarantees its applications over large areas.

I have also checked the code and the data distributed with the paper. The code works perfectly and the data are very good. I really appreciate the effort from the authors to share the code and the data, well done!

However, some points need to be clarified to make the methodology description clearer and to fully understand its applicability in different areas.

On this basis, I believe the paper needs major to moderate changes before the publication; I have listed below my comments with the indication of their relevance.

1) MAJOR: It is not clear if the proposed model is applicable only to permeable pavements or its structure can be generalized and used in different areas. It is an important clarification. Indeed, based on that, several parts of the text should be changed to clarify in which conditions the model can be applied. From the title it seems only for PP, from the text I am confused. Please clarify.

2) MODERATE: A surface layer is introduced into the model structure. It seems very related to its application to permeable pavements, right? If yes, it must be clarified. The acronym FluSM seems to be applicable everywhere, but I am not it is the case. I would appreciate if the authors will discuss the reasons of their choices on the model structure and its parameterization.

3) MAJOR: The parameterization of the developed model is not completely clear to me. The single parameter $I_{cap}$ is estimated trough site specific infiltrometer measurements. I believe this parameter is not easily estimated over large areas and/or over many sites. How to estimate such parameter over several sites? How much is the model sensitive to the value of this parameter? Moreover, in Table 1 I read values of $I_{end}$ higher than 50 mm/h. It means that all rainfall infiltrates into the soil, and hence infiltration plays a very minor role. More details on the estimation of $I_{cap}$ and the model sensitivity to this parameter should be added (in my opinion). On this basis, I would disagree that

input and parameters uncertainties have a small effect on the results, as it reads in the abstract. The model sensitivity to $I_{cap}$ should be included.

4) MINOR: The paper introduces a method for estimate runoff, drainage and evapo-transpiration water fluxes from soil moisture measurements. Other studies have used soil moisture measurement for estimating water fluxes even though mostly specific to a single water flux (e.g., rainfall or evapotranspiration). These studies are not mentioned and should be discussed in the introduction.

**SPECIFIC COMMENT (P: page, L: line or lines)**

P1, L15: "no influx into the soil layer" Which soil layer? Please clarify.

P23, L7-10: For very high infiltration rate, infiltration is not playing a role. In these conditions, all rainfall infiltrates into the soil and fluxes estimation is easier. Please consider this aspect in the discussion.

P24, L9-27: This part is not relevant for the purpose of this paper, particularly L9-15. I would remove this subsection 4.2.

Table 4: The bucket depth shows large variability, e.g., for sites CP2 and GP1 from 96 to 185 mm. How is that possible? Please clarify.

**RECOMMENDATION**

Based on the above comments, I suggest a moderate/major revision before the possible publication on Geoscientific Model Development.

---

## Referee Comment (RC2) · James Ball (Referee) · 16 Nov 2020

In preparing this manuscript, the authors aimed at providing approaches for analysing water fluxes in urban environments and, in particular, those urban environments where permeable pavements existed. For this purpose, a conceptual water balance model is proposed. The contribution to literature arises from the approach used to estimate the conceptual parameter values necessary for operation of the model.

While a continuous modelling system is employed for the water balance model, the time steps employed in the model were 10 minutes for the surface water balance and 1 hour for the remaining components. The response time for most urban surface water

systems is significantly shorter than 1 hour so the approach of using 10-minute computational times is an attempt to mitigate this problem. Unfortunately, in the opinion of this reviewer, a 10-minute computation step is still excessive. The result of the 10-minute computation step will be the absence of computation points on many events and, hence, a lack of data and information contained in the surface flow hydrograph.

For the surface water balance, the two large fluxes will be the precipitation and the evaporation. The surface runoff and infiltration constitute the balancing elements, as noted in the manuscript. Errors in the surface flow hydrographs will be balanced by equal but opposite errors in the infiltration component of the water balance.

The total flux through the surface runoff and infiltration is approximately 900mm. However, the distribution of this 900mm between surface runoff and infiltration is not provided; this data needs to be provided if the 10-minute computation step is to be validated. Furthermore, no surface runoff measurements are provided to validate the assumed values for parameters and the adopted calculation time steps.

The parameter Csurf is defined as the maximum surface storage capacity. As the normal definition of surface storage is the volume of water in temporary transit to the catchment outlet, it is suggested that the term Csurf actually refers to the initial loss storage, sometimes referred to as depression storage; until the depression storage is filled, surface runoff will not be generated which appears to be consistent with the authors' usage of Csurf. It would be interesting if the values obtained using the approach proposed by the authors were similar to the values obtained from the analysis of surface runoff hydrographs; see Boyd et al (1993) and Ball & Powell (1998) for examples of the determination of Csurf from analysis of runoff hydrographs.

While the division between infiltration and surface runoff defined by the authors is consistent with the approach discussed by Rankin & Ball (2010), it is possible for the infiltration capacity to vary with time and thus Icap will not be a constant as assumed by the authors. This variation can occur in both directions. During the storm event,

the infiltration capacity will decrease to an asymptotic rate related to the hydraulic conductivity of the soil; this decrease arises from a decreasing differential between the saturated soil moisture and the antecedent soil moisture. During the dry inter-event periods, there is a recovery of infiltration capacity as the soil moisture decreases.

This variation in infiltration capacity becomes more important as the storm burst duration decreases; in other words, the more prevalent flash events become, the more important it is to assess variations in infiltration capacity. No information about the precipitation events is provided by the authors.

The apparent reliability, i.e. consistency in parameter estimation, of the approach proposed by the authors, maybe related also to the variation in precipitation mechanisms. If there is consistency in the precipitation mechanism and, as noted previously, potential errors are self-compensating, then it is likely that the resultant data clustering would force a convergence of empirical outcomes. Consideration of the rainfall mechanisms and attempting to include a variety of mechanisms would increase confidence in the authors' approach to parameter estimation.

References Ball, JE & Powell, M, (1998), Inference of catchment modelling system control parameters, Proc. UDM '98: Developments in Urban Drainage Modelling, London, UK, Vol. 1, pp 313-320. Boyd, M. J., Bufill, M. C., & Knee, R. M. (1993). Pervious and impervious runoff in urban catchments. Hydrological Sciences Journal, 38(6), 463-478. Rankin, K, & Ball, JE, (2010), The hydrologic performance of permeable pavers, Urban Water, 7(2):79-90.

---

## Referee Comment (RC3) · Anonymous Referee #3 · 3 Dec 2020

Comments on: Fluxes from Soil Moisture Measurements (FluSM v1.0). A Data-driven Water Balance Framework for Permeable Pavements Summary: Interesting presentation of a simple model for infiltration and runoff from permeable surfaces. The model requires much less data than less empirical models. An example calibration with data from a set of permeable pavements was presented. The development of improved methods for design and evaluation of permeable surfaces in urban areas is of rapidly increasing interest as new built urban hardscape is adapted to increase climate resilience with respect to flooding from stormwater, and existing hardscape is being converted to handle need for increased capacity. The continued growth of urban areas increases the significance. The scientific quality appears to be good. The paper is

well referenced, considering the large body of work in the literature. The main limitation appears to be lack of evaluation of the potential significancne of the simplifying assumption that all soil infiltration is vertical, which are posed in the detailed comments below. The science can be reproduced. Recommendations for what the authors think needs to be done to further improve the model, or how to implement the model should be included. The presentation is fine.

Scientific questions: 1. Horizontal permeability can be greater than vertical, and flows can be considerable, especially for PP, because of natural soil deposition which is sheets often which creates horizontal planes of soil fabric with greater permeability, and the inevitable compaction of the subgrade (bottom of the bucket) from construction which reduces vertical infiltration relative to horizontal. Some more discussion about how this might have affected the calibrations. Also, can you give a better idea of horizontal surface area on the bottom of the bucket vs vertical surface area on the sides of the sections used to calibrate to provide an indication how much leaving out the horizontal flow might have affected the calibration results. 2. What was the definition of free draining versus restricted in "Schaffitel et al. (2019) classified the PPs into free-draining PPs"? Please give a one sentence definition 3. Do the case study pavements have a porous reservoir layer? 4. Were there soil hydraulic conductivity measurements for the case study section done prior to installation of the reservoir layer as a check? 5. Pg 24 line 25, Surface permeability is highly variable across a permeable pavement surface at a scale larger than most surface permeability measuring devices.. Generally, not a problem until whole surface clogs because on the same pavement the areas of high permeability areas can handle the flow from low permeability areas nearby. Example: https://www.sciencedirect.com/science/article/pii/S0301479711003525?via%3Dihub Was that also seen in the cited references? 6. Any recommended next steps for this model? Potential improvements? 7. Any issues with extension to include horizontal flow? This can be an important design issue in heavily built environments because of concerns about lateral flow damaging foundations and other infrastructure near the permeable pavement, particularly if it is a street. 8. Could you comment on how you

recommend to implement the model in practice?

Comments on presentation: 1. Pg 2, Line 23, change to "enable the calculation" 2. Pg 3, line 18, change to "lead to an improved" 3. Pg 14, line 4, change "fist" to "first"

---

## Author Comment (AC1) · 3 Dec 2020

**Response to the comments of Anonymous Referee #1 on the manuscript „Fluxes from Soil Moisture Measurements (FluSM v1.0). A Data-driven Water Balance Framework for Permeable Pavements**

We thank Anonymous Referee #1 for reviewing our manuscript, for his positive overall evaluation and for his helpful suggestions for improving the manuscript. In the following, we answer the comments in a point-by-point reply.

**R1C1:** It is not clear if the proposed model is applicable only to permeable pavements or if its structure can be generalized and used in different areas.

> Thank you for this point, which we will clarify in the manuscript. As pointed out in the manuscript, FluSM was designed for fields where the application of Richards based models is critical. Although we applied FluSM only for PPs, FluSM can also be used for different land-use and surface types which we will specify in the following.
>
> For an application of FluSM, soils must fulfill two requirements which are: Drainage must be driven primarily by gravity (due to the used unit gradient approach) and the infiltration capacity must be high (see our answers to R1C3 and R2C7 for an explanation of this requirement).
>
> Besides the application on PPs, FluSM is also applicable for bare soils. Furthermore, we think that FluSM is applicable for sites with vegetation cover, since soil moisture measurements should capture the soil hydrological effect of transpiration. However, for fields with vegetation cover, the location of soil moisture sensors within the profile should be adapted. For our study, we use only shallow measurements since the effect of soil evaporation should be captured best in shallow depths. In contrast, at sites with vegetation cover, root water uptake may act also on deeper depths. Therefore, the installation of multiple sensors covering the entire rooting depth should be considered.
>
> An adaption of the FluSM routine may be necessary e.g. for fields with seasonal varying canopy coverage (e.g. deciduous forests) and sites with seasonal variable vegetation cover (e.g. agricultural fields). Such seasonal changes affect the capacity of the surface storage, which so far is assumed to be constant over time. For a further discussion of the surface storage and possible adaptions see answer to R1C2 and R2C5.
>
> We will include these details in the revised manuscript to clearly define the applicability of the approach.

**R1C2:** It should be clarified whether the surface layer is only related to the application of FluSM to permeable pavements. Model structure and parametrization should be discussed more in detail.

We acknowledge this comment. In the revised manuscript, we will discuss the model structure and parametrization more in detail.

The current implementation of the surface layer assumes that the capacity of the surface storage remains constant over time. This is a reasonable assumption for PPs and should also be valid for e.g. for bare soil and grassland sites. For sites with seasonal varying vegetation/canopy cover, an adaption should be considered to account for the annual variation of the surface storage capacity. As discussed in the manuscript, we think that a seasonal variable surface storage capacity might be determined directly from soil moisture and precipitation data by adapting the method currently implement in FluSM to work on a monthly basis. Therefore, this would require measurements over multiple years. Further alternatives to account for a seasonal variable surface storage capacity include using throughfall measurements or augmenting FluSM by a canopy interception model.

Another characteristic of the surface layer is that the partitioning between infiltration and surface runoff is controlled by the parameter infiltration rate, which remains constant with time. This parameter is discussed in the following point.

**R1C3:** The parameter $I_{cap}$ should be discussed more in detail in terms of model sensitivity, parameter uncertainty and details on the estimation of $I_{cap}$. Additional information on $I_{cap}$ should be provided and a further discussion is expected inter alia in terms of the effect of $I_{cap}$ on uncertainties of the results.

Indeed, estimating the infiltration rate ($I_{cap}$) is crucial, especially for large areas. For PPs, values for $I_{cap}$ in dependence of PP type are provided by Illgen (2009). Since the range of possible $I_{cap}$-values for a given PP-type is high, we recommend using infiltration experiments to derive this parameter site specific. Since FluSM is a data-driven approach which requires plot-specific soil moisture measurements, infiltration experiments could be performed together with the installation of soil moisture sensors. In our study, we used plot-specific $I_{cap}$-values, which were derived from infiltration experiments by Schaffitel et al. (2019). Indeed, those $I_{cap}$-values are quite high (only 5 plots have an $I_{cap}$ < 20 mm/h). However, this is not surprising since constructional requirements call for a high $I_{cap}$ of PPs (FGSV, 2012).

For the reason of parsimony, we use a constant $I_{cap}$ to describe the infiltration process. However, for most soils $I_{cap}$ decreases during the infiltration course, which is mainly due to declining matrix suction gradients during the proceeding of the infiltration front (Hillel, 1998). This is also evident in the data of the infiltration experiments. Thereby, the variability of $I_{cap}$ is documented in a plot-specific infiltration rate derived at the beginning and at the end of the experiment ($I_{start}$ & $I_{end}$) (see Schaffitel et al., 2019).

The results of our uncertainty analysis show that the water balances calculated for PPs with an $I_{cap}$ > 70 mm/h are not sensitive to this parameter. In contrast water balances calculated for 3 PPs with an $I_{cap}$ < 3 mm/h showed a high sensitivity. For a further discussion of $I_{cap}$, we refer to the answer R2C7.

**R1C4:** Additional studies, which have used soil moisture measurements to infer water fluxes, should be discussed in the introduction

We agree, that including further studies which use soil moisture measurements to infer water fluxes will improve the manuscript and will be included in a revised manuscript

**Specific comments**

**P7, L14:** "no influx into the soil layer" Which soil layer? Please clarify.

There is only one single soil layer implemented in FluSM, shown in Fig. 1. To remain consistent with the naming, we will change "soil layer" into "soil storage"

**P23, L7-10:** For very high infiltration rate, surface runoff is not playing a role. In these conditions, all rainfall infiltrates into the soil and fluxes estimation is easier. Please consider this aspect in the discussion.

Indeed, for plots with a high $I_{cap}$, the uncertainty of this parameter has no effect on the water fluxes. We will point out this aspect in our discussion

**P24, L9-27:** This part is not relevant for the purpose of this paper, particularly L9-15. I would remove this subsection 4.2.

Recognized. We will consider removing subsection 4.2

**Table 4:** The bucket depth shows large variability, e.g., for sites CP2 and GP1 from 96 to 185 mm. How is that possible? Please clarify.

Within FluSM, we use a regression approach to derive the bucket depth from the observed soil moisture reaction and from the infiltration calculated by the surface water balance. Since surface runoff is negligible on plots CP2 and GP1 (both plots have a very high infiltration rate), the amount of total infiltration should be comparable (although not identical e.g. due to different surface storage capacities). Hence, the deviations between the derived bucket depths originate from differences in the amplitude of the soil moisture reaction to infiltration. Such may be caused by differences in the 3-dimensional propagation of the wetting front underneath the joints (e.g. caused by spatial distribution of impermeable paving stones and joints or by differences in soil properties), soil-specific parameters (e.g. the amount of skeleton) and by the connection of soil moisture sensors to surrounding soils. Due to the derivation of the bucket depth by a regression approach, all site-specific characteristics are lumped in this parameter which hampers a physical interpretation.

**Literature**

FGSV: Richtlinien für die Standardisierung des Oberbaus., 2012.

Hillel, D.: Environmental Soil Physics, Acad. Press, San Diego, Calif., 1998.

Illgen, M.: Das Versickerungsverhalten durchlässig befestigter Siedlungsflächen und seine urbanhydrologische Quantifizierung., 2009.

Schaffitel, A., Schuetz, T. and Weiler, M.: A distributed soil moisture, temperature and infiltrometer dataset for permeable pavements and green spaces, Earth Syst. Sci. Data Discuss., 1–27, doi:10.5194/essd-2019-97, 2019.

---

## Author Comment (AC2) · 3 Dec 2020

**Response to the comments of Referee #2 (James Ball) on the manuscript „Fluxes from Soil Moisture Measurements (FluSM v1.0). A Data-driven Water Balance Framework for Permeable Pavements"**

We thank James Ball for his general remarks, his comments on urban hydrology and the points concerning the structure of FluSM. In the revision, we will consider those points which will help us improving the quality of the manuscript. We are deeply grateful for that.

**R2C1**: The response time for most urban surface water systems is significantly shorter than 10 min. Using a 10-minute computation step results in a lack of information and data in the surface flow hydrograph

> A key characteristic of urban areas is the fast concentration, collection and conveyance of surface runoff (Shuster et al., 2005). This causes a high flashiness in surface flow hydrographs and modelling calls for a high temporal resolution of rainfall data. Besides the temporal resolution, also the spatial resolution of rainfall is decisive, since urban hydrological processes are characterized by a high variability not only in time, but also in space (Cristiano et al., 2017). Since high resolution rainfall data is rarely available, precipitation is often seen as a main source of uncertainty in urban hydrology (Cristiano et al., 2017; Niemczynowicz, 1999). This might also be the case for our study, for which we used rainfall data with a 10-min temporal resolution originating from one single urban climate station. Due to the location of our study sites within the public urban space, it was not possible to set-up site-specific rainfall gauges. We are aware that both factors (the spatial location of precipitation measurements and the temporal resolution) lead to an uncertainty of the precipitation input used for our study. However, we accounted for this uncertainty within our uncertainty analysis.
>
> Within the uncertainty analysis, we accounted for the spatial heterogeneity of rainfall by using time series of different climate stations as ensembles. In order to account for small-scale rainfall variability, we additionally multiplied the time series by a factor ranging between 0.8 and 1.2. By doing so, we considered a large uncertainty range for precipitation (550-1150 mm/year), which we think should also account for the uncertainty caused by the 10-min temporal resolution. The results of the uncertainty analysis reveal that the effect on surface runoff is small for most plots. Only the results for 3 plots (GP15, CP14 and CP13), show large uncertainties in surface runoff, which we attribute to the low infiltration rate of those plots. However, the uncertainty of the results obtained for those plots, is also caused by the input uncertainty of precipitation. We will clarify this in the manuscript. Furthermore, we will point out that the effect of uncertain precipitation input depends on the plot-specific infiltration rate.

**R2C2**: Errors in the surface flow hydrographs will be balanced by equal but opposite errors in the infiltration component of the water balance (extension of R2C1)

Indeed, errors in the calculated infiltration lead to opposite errors which are equal in absolute value in surface runoff. Uncertainties in precipitation and in the infiltration rate may cause such an error in infiltration and surface runoff. Its possible magnitude is reflected in the uncertainty ranges obtained for surface runoff (Fig. 11). The results show that this error is negligible for plots with an infiltration rate above 70 mm/h while it is high for plots with an infiltration rate below 3 mm/h (CP15, CP14 and CP13). We will clarify this point in the revised manuscript.

**R2C3**: The distribution between surface runoff and infiltration needs to be provided if the 10-minute computation step is to be validated

Currently, FluSM returns time series for all water fluxes with a temporal resolution of 1 h. We will adapt the code of FluSM in the way that the surface water balance will be returned with a temporal resolution of 10 min.

**R2C4**: Surface runoff measurements are not provided for validation

We agree that such measurements would be desirable for validation. Most valuable would be measurements at the plot scale, since runoff measurements integrating large areas (e.g. measurements in sewer drains) would be difficult to interpret for the plot scale. Unfortunately, our plots are located in the public urban space (e.g. on residential roads, bicycle tracks, parking lots and pedestrian roads) and we are not aware of any practicable and affordable measurement set-up, suited for continuously measuring plot-scale surface runoff within the public urban space. Due to this, such measurements do not exist. However, there is data for plot-specific infiltration experiments provided in Schaffitel et al. (2019).

**R2C5**: $C_{surf}$ is defined as the surface storage capacity, which is normally defined as the volume of water in temporary transit to the catchment outlet. It is therefore suggested that $C_{surf}$ refers to the initial loss storage (sometimes also referred to as depression storage)

In FluSM, the surface storage is the water storage exiting at the atmosphere-soil/pavement boundary. Following Mansell & Rollet (2009) the surface storage consists of a depression storage (storage due to the micro relief of the surface) and the wetting capacity of the surface (amount of water required for wetting the surface).

To our knowledge, in urban hydrology, the initial loss is often determined by a linear regression of runoff against rainfall (intersect with the x-axis; e.g. Rodriguez et al., 2000).

For sake of clarity, we decided to use the term surface storage instead of initial loss. Furthermore, we decided to clearly distinguish between the state of this storage ($S_{surf}$) and its capacity ($C_{surf}$).

**R2C6**: A comparison of the obtained $C_{surf}$ values with initial loss obtained by previous studies would be interesting

> Indeed, we think that such a comparison would be valuable for the manuscript and we will include this in our discussion.

**R2C7:** The parameter $I_{cap}$ may vary with time and not be a constant as assumed by the authors. Temporal distribution of storm and inter-storm periods determines the variability of $I_{cap}$ and therefore additional information about precipitation events and mechanisms should be provided. In case of consistent precipitation mechanisms and self-compensating errors, the results could be reliable. Consideration of the rainfall mechanisms and attempting to include a variety of mechanisms would increase confidence in the authors' approach to parameter estimation.

> This is a very interesting point. As pointed out in R1C3 and R2C2, we will further discus the parameter $I_{cap}$ in the manuscript. However, we think that an additional analysis of precipitation events and mechanisms will not lead to further insights, which we will explain in the following.
>
> We agree that the infiltration rate may vary with time, which is caused by a change of soil moisture during the infiltration course (which in turn controls the matrix potential and the hydraulic conductivity). However, describing infiltration only by matrix flux might be insufficient for PPs, since infiltration might be controlled also by other processes (e.g. preferential flow and hydrophobicity). For our plots, the variability of the infiltration rate over time is documented by plot-specific infiltration experiments under ponded conditions (see Schaffitel et al., 2019). Those experiments were used to derive a plot-specific infiltration rate for the beginning and for the end of the infiltration course ($I_{start}$ & $I_{end}$). Thereby, $I_{start}$ represents the infiltration rate when soils are dry, while $I_{end}$ represents infiltration under steady-state conditions (constant soil moisture, matrix potential and hydraulic conductivity). Hence, the documented $I_{start}$ and $I_{end}$ should capture the possible variability of the infiltration rate caused by the temporal distribution of storm and inter-storm periods. We considered this variability in our uncertainty analysis and discussed its effect on the water balance. Thereby, the results show that the uncertainty of the parameter $I_{cap}$ (and hence also the effect of the temporal storm and inter-storm distribution on this parameter) is relevant only for 3 plots with a very low $I_{cap}$, while it is negligible for the majority of the plots. Due to this, the results of FluSM for plots with a low $I_{cap}$ should be regarded with care, while results are reliable for plots with an $I_{cap}$ of at least 9 mm/h. In the revised manuscript, we will put a stronger emphasize on the requirement of high $I_{cap}$-values for the reliability of FluSM.

**Literature**

Cristiano, E., Veldhuis, M. C. Ten and Van De Giesen, N.: Spatial and temporal variability of rainfall and their effects on hydrological response in urban areas - A review, Hydrol. Earth Syst. Sci., 21(7), 3859–3878, doi:10.5194/hess-21-3859-2017, 2017.

Mansell, M. and Rollet, F.: The effect of surface texture on evaporation, infiltration and storage properties of paved surfaces, Water Sci. Technol., 60(1), 71–76, doi:10.2166/wst.2009.323, 2009.

Niemczynowicz, J.: Urban hydrology and water management – present and future challenges, Urban Water, 1(1), 1–14, doi:10.1016/S1462-0758(99)00009-6, 1999.

Rodriguez, F., Andrieu, H. and Zech, Y.: Evaluation of a distributed model for urban catchments using a 7-year continuous data series, Hydrol. Process., 14(5), 899–914, doi:10.1002/(SICI)1099-1085(20000415)14:5<899::AID-HYP977>3.0.CO;2-R, 2000.

Schaffitel, A., Schuetz, T. and Weiler, M.: A distributed soil moisture, temperature and infiltrometer dataset for permeable pavements and green spaces, Earth Syst. Sci. Data Discuss., 1–27, doi:10.5194/essd-2019-97, 2019.

Shuster, W. D., Bonta, J., Thurston, H., Warnemuende, E. and Smith, D. R.: Impacts of impervious surface on watershed hydrology: A review, Urban Water J., 2(4), 263–275, doi:10.1080/15730620500386529, 2005.

---

## Author Comment (AC3) · 2 Jan 2021

**Response to the comments of Anonymous Referee #3 on the manuscript „Fluxes from Soil Moisture Measurements (FluSM v1.0). A Data-driven Water Balance Framework for Permeable Pavements**

We thank Anonymous Referee #3 for the positive feedback and for the constructive comments, which we will consider to improve the manuscript. In the following, we answer the comments in a point-by-point reply.

**R3C1:** Horizontal permeability can be greater than vertical, and flows can be considerable, especially for PP, because of natural soil deposition which is sheets often which creates horizontal planes of soil fabric with greater permeability, and the inevitable compaction of the subgrade (bottom of the bucket) from construction which reduces vertical infiltration relative to horizontal. Some more discussion about how this might have affected the calibrations. Also, can you give a better idea of horizontal surface area on the bottom of the bucket vs vertical surface area on the sides of the sections used to calibrate to provide an indication how much leaving out the horizontal flow might have affected the calibration results.

> Indeed, horizontal subsurface flow may account for a large share of the water balance for PPs, especially since the hydraulic conductivity of underlying soil layers may be low due to the compaction of the subgrade. In the following, we will explain, why we think that horizontal subsurface flow did not affect the calibration of FluSM for the PPs of the case study.
>
> For the layers of a PP (pavement, bedding, base and subbase layer), a high hydraulic conductivity is required. Therefore, vertical flow should dominate within those layers. If the conductivity of underlying soil layers is low, there are two possible cases. Either horizontal subsurface flow occurs at the bottom of those layers, or the soil storage gets filled gradually until there is saturation overland flow. Since we applied FluSM only for PPs with a "free drainage behavior" (see R3C2), the second case is not relevant for our study.
>
> Within FluSM, the observed soil moisture recession is used to calibrate a simple drainage model. For this calibration, it is not relevant whether the soil moisture recession is due to vertical drainage or if it is due to horizontal subsurface flow. Both fluxes are summarized in the calculated drainage flux. The calibration of FluSM would be problematic only for PPs showing a "restricted drainage behavior", which were therefore excluded from our study.
>
> Regarding a possible separation between vertical and horizontal subsurface flow, we refer to our answer on R3C7.

**R3C2**: What was the definition of free draining versus restricted in "Schaffitel et al. (2019) classified the PPs into free-draining PPs"? Please give a one sentence definition

Thank you for pointing out the missing definition. We will clarify this by adding the following description to the revised manuscript:

The classification applied in Schaffitel et al. (2019), is based on a combination of statistical analysis and visual inspection. Plots were classified as "restricted drainage" when soil moisture reached saturation frequently during rain events and remained saturated even after the end of rainfall. In contrast, plots which showed a fast recession of soil moisture were classified as "free drainage". Thereby, the fast soil moisture recession indicates a high hydraulic conductivity of underlying soil layers.

**R3C3:** Do the case study pavements have a porous reservoir layer?

According to the local construction authority, the PPs of the case study should not have a porous reservoir layer. Except two plots, this is in accordance with the observations made during field works. Only at two plots (CP12 and CP2) we encountered coarse gravel underneath the pavement layer which could serve as kind of porous reservoir layer. Those two plots are located on a private parking lot.

We will clarify this in the revised manuscript.

**R3C4:** Were there soil hydraulic conductivity measurements for the case study section done prior to installation of the reservoir layer as a check?

This is an interesting question. Unfortunately, information is neither available for construction works, nor for preceding measurements. We therefore planned to extract undisturbed soil samples for determining the hydraulic conductivity function from multistep-outflow experiments in the laboratory. Due to the high fraction of soil skeleton and due to the high soil compaction it was impossible to extract undisturbed soil samples. However, for the PPs of our study the soil hydraulic conductivity of underlying soil layers should be high since all PPs were classified as "free drainage" (see R3C").

**R3C5:** Pg 24 line 25, Surface permeability is highly variable across a permeable pavement surface at a scale larger than most surface permeability measuring devices. Generally, not a problem until whole surface clogs because on the same pavement the areas of high permeability areas can handle the flow from low permeability areas nearby. Example: https://www.sciencedirect.com/science/article/pii/S0301479711003525?via%3Dihub Was that also seen in the cited references?

Indeed, there are various studies showing the variability of surface clogging across a permeable pavement surface (e.g. Razzaghmanesh and Beecham, 2018; Sañudo-Fontaneda et al., 2014). Factors controlling the surface clogging of PPs include age, traffic load, maintenance measures, surrounding land use, joint proportion and filling material of joints (Boogaard et al., 2014a; Winston et al., 2016). Previous studies showed, that surface clogging occurs mainly in the first years after the construction (e.g. Boogaard et al., 2014b; Borgwardt, 2006; Lucke and Beecham, 2011). The effect of run-on from surrounding surfaces on PP clogging was investigated e.g. by Razzaghmanesh and Borst (2018).

At the plots of our study, infiltration experiments were performed only once at the beginning of the study period. Due to the lack of successive infiltration experiments, a direct quantification of the clogging progress over the study period is not possible. However, soil moisture time series should allow for an indirect assessment, since surface clogging affects the infiltration capacity which in turn affects soil moisture dynamics. In this way, Razzaghmanesh and Borst (2018), used soil moisture measurements to study clogging dynamics of a PP surface.

For the PPs of our study, we analyzed the measured soil moisture dynamics over the study period. Since we did not observe a change in dynamics over time, we expect that the state of surface clogging remained more or less constant over the study period. One possible explanation therefore might be that none of the PPs was newly build and therefore all plots were already clogged at the beginning of the study period.

We will clarify this in the revised manuscript, accordingly

**R3C6:** Any recommended next steps for FluSM and potential improvements

Thank you for this comment. We will discuss next steps and possible improvements in more detail in the revised manuscript.

Potential improvements include adaptions for the application on sites with vegetation cover and the consideration of horizontal subsurface flow. For a detailed discussion on those improvements we refer to our answers on R1C1 and R3C7. Concerning recommendations for next steps, we refer to the answer on R3C8.

**R3C7:** Possibility to extend FluSM to account also for horizontal flow, which might be important for estimating possible effects on surrounding infrastructure

Indeed, this is a very interesting point. In the following, we will point out a parsimonious concept which allows extending FluSM to account also for horizontal subsurface flow on PPs.

For PPs, the occurrence of horizontal subsurface flow is mainly limited to the border subbase layer – underlying soil, since this border might be associated with a strong decrease in soil hydraulic conductivities. Describing horizontal subsurface flow at this border requires knowledge on the soil hydraulic parameters of both layers. In a parsimonious approach, the saturated hydraulic conductivity of the underlying soil layer could be used as single parameter to describe the partitioning between deep percolation and horizontal subsurface flow at this border. Thereby, the saturated hydraulic conductivity needs to be determined e.g. during the construction of PPs.

We think that such an extension is beyond the scope of this paper, since we applied FluSM only for PPs showing a free drainage behavior. However, in the revised manuscript, we will discuss the aforementioned possibility for an extension.

**R3C8:** Recommendations to implement the model in practice

Thank you for this remark. We will include the following recommendations in the discussion.

The main advantage of FluSM is the possibility to derive continuous water fluxes from soil moisture and meteorological measurements in a relative easy and cheap way. Therefore, FluSM allows to study the water balance of fields with limited knowledge (e.g. missing soil hydrologic parameters or lack of knowledge on the correct representation of processes). Regarding the ever-increasing availability of soil moisture data on different spatial scales, the demand of such parsimonious approaches should increase.

So far, long-term, high resolution hydrological fluxes of PPs under field conditions were obtained only by lysimeter studies. Since such measurements are costly, the availability of data for validating generalized modelling approaches is limited. In the future, data-driven derivations of soil hydrological fluxes might serve as a simulation benchmark for the application of process based urban hydrological models.

Comments on presentation
**P2, L23**: change to "enable the calculation"

Acknowledged

**P2, L18:** change to "lead to an improved"

Acknowledged

**P14, L4:** change "fist" to "first"

Acknowledged

**Literature**

Boogaard, F., Lucke, T. and Beecham, S.: Effect of age of permeable pavements on their infiltration function, Clean - Soil, Air, Water, 42(2), 146–152, doi:10.1002/clen.201300113, 2014a.

Boogaard, F., Lucke, T., van de Giesen, N. and van de Ven, F.: Evaluating the infiltration performance of eight dutch permeable pavements using a new full-scale infiltration testing method, Water (Switzerland), 6(7), 2070–2083, doi:10.3390/w6072070, 2014b.

Borgwardt, S.: Long-Term In-Situ Infiltration Performance of Permeable Concrete Block Pavement, 8th Int. Conf. Concr. Block Paving, Novemb. 6-8, 2006 San Fr. Calif. USAnternational Conf. Concr. block paving, 149–160 [online] Available from: http://citeseerx.ist.psu.edu/viewdoc/download?doi=10.1.1.365.9174&rep=rep1&type=pdf, 2006.

Lucke, T. and Beecham, S.: Field investigation of clogging in a permeable pavement system, Build. Res. Inf., 39(6), 603–615, doi:10.1080/09613218.2011.602182, 2011.

Razzaghmanesh, M. and Beecham, S.: A review of permeable pavement clogging investigations and recommended maintenance regimes, Water (Switzerland), 10(3), doi:10.3390/w10030337, 2018.

Razzaghmanesh, M. and Borst, M.: Investigation clogging dynamic of permeable pavement systems using embedded sensors, J. Hydrol., 557, 887–896, doi:10.1016/j.jhydrol.2018.01.012, 2018.

Sañudo-Fontaneda, L. A., Andrés-Valeri, V. C. A., Rodriguez-Hernandez, J. and Castro-Fresno, D.: Field study of infiltration capacity reduction of porous mixture surfaces, Water (Switzerland), 6(3), 661–669, doi:10.3390/w6030661, 2014.

Schaffitel, A., Schuetz, T. and Weiler, M.: A distributed soil moisture, temperature and infiltrometer dataset for permeable pavements and green spaces, Earth Syst. Sci. Data Discuss., 1–27, doi:10.5194/essd-2019-97, 2019.

Winston, R. J., Al-Rubaei, A. M., Blecken, G. T., Viklander, M. and Hunt, W. F.: Maintenance measures for preservation and recovery of permeable pavement surface infiltration rate - The effects of street sweeping, vacuum cleaning, high pressure washing, and milling, J. Environ. Manage., 169, 132–144, doi:10.1016/j.jenvman.2015.12.026, 2016.

---

## Author Response (AR2)

Dear editor,

Thank you for editing our manuscript. The first part of this document includes the point-by-point response to the reviews (R1, R2, R3). Comments of the referees are marked as e.g. << R1C1: "referees' comment">> followed by the answer from the authors, which includes a description of the changes made in the manuscript to fulfill the referees' suggestions.

Best regards,
Schaffitel et al.

**Response to the comments of Anonymous Referee #1 on the manuscript „Fluxes from Soil Moisture Measurements (FluSM v1.0). A Data-driven Water Balance Framework for Permeable Pavements**

We thank Anonymous Referee #1 for reviewing our manuscript, for his positive overall evaluation and for his helpful suggestions for improving the manuscript. In the following, we answer the comments in a point-by-point reply.

**R1C1:** It is not clear if the proposed model is applicable only to permeable pavements or if its structure can be generalized and used in different areas.

> Thank you for this point, which we clarified in the manuscript. As pointed out in the manuscript, FluSM was designed for fields where the application of Richards based models is critical. Although we applied FluSM only for PPs, FluSM can also be used for different land-use and surface types specified in the following.
>
> For an application of FluSM, soils must fulfill two requirements which are: Drainage must be driven primarily by gravity (due to the used unit gradient approach) and the infiltration capacity must be high (see our answers to R1C3 and R2C7 for an explanation of this requirement).
>
> Besides the application on PPs, FluSM is also applicable for bare soils. Furthermore, we think that FluSM is applicable for sites with vegetation cover, since soil moisture measurements should capture the soil hydrological effect of transpiration. However, for sites with vegetation cover, the location of soil moisture sensors within the profile should be adapted. For our study, we use only shallow measurements since the effect of soil evaporation should be captured best in shallow depths. In contrast, at sites with vegetation cover, root water uptake may act also on deeper depths. Therefore, the installation of multiple sensors covering the entire rooting depth should be considered.
>
> An adaption of the FluSM routine may be necessary e.g. for fields with seasonal varying canopy coverage (e.g. deciduous forests) and sites with seasonal variable vegetation cover (e.g. agricultural fields). Such seasonal changes affect the capacity of the surface storage, which so far is assumed to be constant over time. For a further discussion of the surface storage and possible adaptions see our answer to R1C2 and R2C5.
>
> We included these details in the discussion of the revised manuscript to clearly define the applicability of the approach.

**R1C2:** It should be clarified whether the surface layer is only related to the application of FluSM to permeable pavements. Model structure and parametrization should be discussed more in detail.

> In the revised manuscript this point is acknowledged by the following discussion:
>
> Currently, the implementation assumes that the surface storage capacity remains constant over time. This is a reasonable assumption for PPs and should also be valid for e.g. for bare soil and grassland sites. However, for sites with seasonal varying vegetation/canopy cover the concept of a constant surface storage capacity might be problematic due to the seasonally variable canopy coverage (Link et al., 2004). Under such conditions, an adaption of the routine should be considered. We think that a seasonal variable surface storage capacity might be determined directly from soil moisture and precipitation data by adapting the determination (FluSM step 1) to work on a monthly basis. This would require measurements over multiple years. Further alternatives to account for a seasonal variable surface storage capacity include using throughfall measurements or augmenting FluSM by a canopy interception model.
>
> Another characteristic of the surface layer is that the partitioning between infiltration and surface runoff is controlled by the parameter infiltration rate, which remains constant with time. This parameter is discussed in R1C3.

**R1C3:** The parameter $I_{cap}$ should be discussed more in detail in terms of model sensitivity, parameter uncertainty and details on the estimation of $I_{cap}$. Additional information on $I_{cap}$ should be provided and a further discussion is expected inter alia in terms of the effect of $I_{cap}$ on uncertainties of the results.

> Indeed, estimating the infiltration rate ($I_{cap}$) is crucial, especially for large areas. For PPs, values for $I_{cap}$ in dependence of PP type are provided by Illgen (2009). Since the range of possible $I_{cap}$-values for a given PP-type is high, we recommend using infiltration experiments to derive this parameter site-specifically. Since FluSM is a data-driven approach which requires plot-specific soil moisture measurements, infiltration experiments could be performed together with the installation of soil moisture sensors. In our study, we used plot-specific $I_{cap}$-values, which were derived from infiltration experiments by Schaffitel et al. (2019). Indeed, those $I_{cap}$-values are quite high (only 5 plots have an $I_{cap}$ < 20 mm/h). However, this is not surprising since constructional requirements call for a high $I_{cap}$ of PPs (FGSV, 2012).
>
> For the reason of parsimony, we use a constant $I_{cap}$ to describe the infiltration process. However, for most soils $I_{cap}$ decreases during the infiltration course, which is mainly due to declining matrix suction gradients during the proceeding of the infiltration front (Hillel, 1998). This is also evident in the data of the infiltration experiments. Thereby, the variability of $I_{cap}$ is documented in a plot-specific infiltration rate derived at the beginning and at the end of the experiment ($I_{start}$ & $I_{end}$) (see Schaffitel et al., 2019).

The results of our uncertainty analysis show that the water balances calculated for PPs with an $I_{cap}$ > 70 mm/h are not sensitive to this parameter. In contrast water balances calculated for 3 PPs with an $I_{cap}$ < 3 mm/h showed a high sensitivity. For a further discussion of $I_{cap}$, we refer to the answer R2C7.

We clarified these points in the revised manuscript

**R1C4:** Additional studies, which have used soil moisture measurements to infer water fluxes, should be discussed in the introduction

We added the following paragraph to the introduction which points out a common way of using soil moisture measurements for soil hydrologic modelling.

One possibility for using soil moisture measurements for vadose zone modelling is by using them to determine soil hydrologic properties inversely (e.g. Ries et al., 2015; Ritter et al., 2003; Wollschläger et al., 2009). Nevertheless, estimating soil hydrological parameters from soil moisture data only leads to equifinality of parameter sets. Hence, additional information should be incorporated into the inverse estimation procedure to further constrain the obtained parameters (Vereecken et al., 2010).

**Specific comments**
P7, L14: "no influx into the soil layer" Which soil layer? Please clarify.

There is only one single soil layer implemented in FluSM, shown in Fig. 1. To remain consistent with the naming, we changed "soil layer" into "soil storage"

**P23, L7-10:** For very high infiltration rate, surface runoff is not playing a role. In these conditions, all rainfall infiltrates into the soil and fluxes estimation is easier. Please consider this aspect in the discussion.

Indeed, for plots with a high $I_{cap}$, the uncertainty of this parameter has no effect on the water fluxes. We considered this aspect by adding the following point to our discussion: *"…This highlights the requirement of a high plot-specific infiltration rate for the reliability of the FluSM approach…".*

**P24, L9-27:** This part is not relevant for the purpose of this paper, particularly L9-15. I would remove this subsection 4.2.

We removed subsection 4.2 from the manuscript

**Table 4:** The bucket depth shows large variability, e.g., for sites CP2 and GP1 from 96 to 185 mm. How is that possible? Please clarify.

Within FluSM, we use a regression approach to derive the bucket depth from the observed soil moisture reaction and from the infiltration calculated by the surface water balance. Since surface runoff is negligible on plots CP2 and GP1 (both plots have a very high infiltration rate), the amount of total infiltration should be comparable (although not identical e.g. due to different surface storage capacities). Hence, the deviations between the derived bucket depths originate from differences in the amplitude of the soil moisture reaction to infiltration. Such may be caused by differences in the 3-dimensional propagation of the wetting front underneath the joints (e.g. caused by spatial distribution of impermeable paving stones and joints or by differences in soil properties), soil-specific parameters (e.g. the amount of skeleton) and by the connection of soil moisture sensors to surrounding soils. Due to the derivation of the bucket depth by a regression approach, all site-specific characteristics are lumped in this parameter which hampers a physical interpretation.

We clarified this point in the revised manuscript accordingly.

**Response to the comments of Referee #2 (James Ball) on the manuscript „Fluxes from Soil Moisture Measurements (FluSM v1.0). A Data-driven Water Balance Framework for Permeable Pavements"**

We thank James Ball for his general remarks, his comments on urban hydrology and the points concerning the structure of FluSM. In the revision, we will consider those points which will help us improving the quality of the manuscript. We are deeply grateful for that.

**R2C1**: The response time for most urban surface water systems is significantly shorter than 10 min. Using a 10-minute computation step results in a lack of information and data in the surface flow hydrograph

Thank you for this comment, which we considered in the revised manuscript by discussing the following points:

A key characteristic of urban areas is the fast concentration, collection and conveyance of surface runoff (Shuster et al., 2005). This causes a high flashiness in surface flow hydrographs and modelling calls for a high temporal resolution of rainfall data. Besides the temporal resolution, also the spatial resolution of rainfall is decisive, since urban hydrological processes are characterized by a high variability not only in time, but also in space (Cristiano et al., 2017). Since high resolution rainfall data is rarely available, precipitation is often seen as a main source of uncertainty in urban hydrology (Cristiano et al., 2017; Niemczynowicz, 1999). This might also be the case for our study, for which we used rainfall data with a 10-min temporal resolution originating from one single urban climate station. Due to the location of our study sites within the public urban space, it was not possible to set-up site-specific rainfall gauges. We are aware that both factors (the spatial location of precipitation measurements and the temporal resolution) lead to an uncertainty of the precipitation input used for our study. However, we accounted for this uncertainty within our uncertainty analysis.

Within the uncertainty analysis, we accounted for the spatial heterogeneity of rainfall by using time series of different climate stations as ensembles. In order to account for small-scale rainfall variability, we additionally multiplied the time series by a factor ranging between 0.8 and 1.2. By doing so, we considered a large uncertainty range for precipitation (550-1150 mm/year), which we think should also account for the uncertainty caused by the 10-min temporal resolution. The results of the uncertainty analysis reveal that the effect on surface runoff is small for most plots. Only the results for 3 plots (GP15, CP14 and CP13), show large uncertainties in surface runoff, which we attribute to the low infiltration rate of those plots. However, the uncertainty of the results obtained for those plots, is also caused by the input uncertainty of precipitation.

**R2C2**: Errors in the surface flow hydrographs will be balanced by equal but opposite errors in the infiltration component of the water balance (extension of R2C1)

Indeed, errors in the calculated infiltration lead to opposite errors which are equal in absolute value in surface runoff. Uncertainties in precipitation and in the infiltration rate may cause such an error in infiltration and surface runoff. Its possible magnitude is reflected in the uncertainty ranges obtained for surface runoff (Fig. 11). The results show that this error is negligible for plots with an infiltration rate above 70 mm/h while it is high for plots with an infiltration rate below 3 mm/h (CP15, CP14 and CP13).

We clarified this point in the revised manuscript.

**R2C3**: The distribution between surface runoff and infiltration needs to be provided if the 10-minute computation step is to be validated

We updated the code of FluSM on the Gitlab repository (https://gitlab.com/ASchaffitel/flusm). Now, the surface water balance is returned also with a temporal resolution of 10 min.

**R2C4**: Surface runoff measurements are not provided for validation

We agree that such measurements would be desirable for validation. Most valuable would be measurements at the plot scale, since runoff measurements integrating large areas (e.g. measurements in sewer drains) would be difficult to interpret for the plot scale. Unfortunately, our plots are located in the public urban space (e.g. on residential roads, bicycle tracks, parking lots and pedestrian roads) and we are not aware of any practicable and affordable measurement set-up, suited for continuously measuring plot-scale surface runoff within the public urban space. Due to this, such measurements do not exist. However, there is data for plot-specific infiltration experiments provided in Schaffitel et al. (2019).

**R2C5**: $C_{surf}$ is defined as the surface storage capacity, which is normally defined as the volume of water in temporary transit to the catchment outlet. It is therefore suggested that $C_{surf}$ refers to the initial loss storage (sometimes also referred to as depression storage)

To our knowledge, in urban hydrology, the initial loss is often determined by a linear regression of runoff against rainfall (intersect with the x-axis; e.g. Rodriguez et al., 2000). For sake of clarity, we decided to use the term surface storage instead of initial loss. Furthermore, we decided to clearly distinguish between the state of this storage ($S_{surf}$) and its capacity ($C_{surf}$).

To clarify this point, we included the following description to the manuscript: In FluSM, the surface storage is the water storage exiting at the atmosphere-soil/pavement boundary. Following Mansell & Rollet (2009) the surface storage consists of a depression storage (storage due to the micro relief of the surface) and the wetting capacity of the surface (amount of water required for wetting the surface).

**R2C6**: A comparison of the obtained $C_{surf}$ values with initial loss obtained by previous studies would be interesting

Indeed, we think that such a comparison is valuable for the manuscript and we therefore added the following paragraph to our discussion:

Regarding the surface storage capacity of PPs, we obtained values between 1 mm and 4.5 mm, which is in accordance with the range specified by previous studies (e.g. Brown and Borst, 2015; Flöter, 2006; Illgen, 2009; Starke et al., 2010; Wessolek et al., 2008; Wessolek and Facklam, 1997; Wiles and Sharp, 2008).

**R2C7:** The parameter $I_{cap}$ may vary with time and not be a constant as assumed by the authors. Temporal distribution of storm and inter-storm periods determines the variability of $I_{cap}$ and therefore additional information about precipitation events and mechanisms should be provided. In case of consistent precipitation mechanisms and self-compensating errors, the results could be reliable. Consideration of the rainfall mechanisms and attempting to include a variety of mechanisms would increase confidence in the authors' approach to parameter estimation.

This is a very interesting point. As pointed out in R1C3 and R2C2, we added an additional discussion of the parameter $I_{cap}$ to the manuscript. However, we think that an additional analysis of precipitation events and mechanisms will not lead to further insights, which we will explain in the following.

We agree that the infiltration rate may vary with time, which is caused by a change of soil moisture during the infiltration course (which in turn controls the matrix potential and the hydraulic conductivity). However, describing infiltration only by matrix flux might be insufficient for PPs, since infiltration might be controlled also by other processes (e.g. preferential flow and hydrophobicity). For our plots, the variability of the infiltration rate over time is documented by plot-specific infiltration experiments under ponded conditions (see Schaffitel et al., 2019). Those experiments were used to derive a plot-specific infiltration rate for the beginning and for the end of the infiltration course ($I_{start}$ & $I_{end}$). Thereby, $I_{start}$ represents the infiltration rate when soils are dry, while $I_{end}$ represents infiltration under steady-state conditions (constant soil moisture, matrix potential and hydraulic conductivity). Hence, the documented $I_{start}$ and $I_{end}$ should capture the possible variability of the infiltration rate caused by the temporal distribution of storm and inter-storm periods. We considered this variability in our uncertainty analysis and discussed its effect on the water balance. Thereby, the results show that the uncertainty of the parameter $I_{cap}$ (and hence also the effect of the temporal storm and inter-storm distribution on this parameter) is relevant only for 3 plots with a very low $I_{cap}$, while it is negligible for the majority of the plots. Due to this, the results of FluSM for plots with a low $I_{cap}$ should be regarded with care, while results are reliable for plots with an $I_{cap}$ of at least 9 mm/h.

In the revised manuscript, we put a stronger emphasize on the requirement of high $I_{cap}$-values for the reliability of FluSM.

**Response to the comments of Anonymous Referee #3 on the manuscript „Fluxes from Soil Moisture Measurements (FluSM v1.0). A Data-driven Water Balance Framework for Permeable Pavements**

We thank Anonymous Referee #3 for the positive feedback and for the constructive comments, which we will consider to improve the manuscript. In the following, we answer the comments in a point-by-point reply.

**R3C1:** Horizontal permeability can be greater than vertical, and flows can be considerable, especially for PP, because of natural soil deposition which is sheets often which creates horizontal planes of soil fabric with greater permeability, and the inevitable compaction of the subgrade (bottom of the bucket) from construction which reduces vertical infiltration relative to horizontal. Some more discussion about how this might have affected the calibrations. Also, can you give a better idea of horizontal surface area on the bottom of the bucket vs vertical surface area on the sides of the sections used to calibrate to provide an indication how much leaving out the horizontal flow might have affected the calibration results.

Indeed, horizontal subsurface flow may account for a large share of the water balance for PPs, especially since the hydraulic conductivity of underlying soil layers may be low due to the compaction of the subgrade. In the following we explain why horizontal subsurface flow should not affect the calibration of the drainage model. We clarified this accordingly in the revised manuscript.

Within a PP system (pavement, bedding, base and subbase layer), horizontal subsurface flow should play an only minor role, since a high hydraulic conductivity is required for all layers. This is also reflected in the free drainage behavior observed for the PPs of our case study. However, horizontal subsurface flow may occur at the bottom of the PP system (e.g. border between subbase layer and underlying soil). Such horizontal subsurface flow would not affect the calibration of the drainage model, since it is not it is not relevant whether the soil moisture recession is due to vertical drainage or if it is due to horizontal subsurface flow. Both fluxes are summarized in the calculated drainage flux. The calibration of FluSM would be problematic only for plots showing a restricted drainage behavior, which we therefore excluded from our case study

Regarding a possible separation between vertical and horizontal subsurface flow, we refer to our answer on R3C7.

**R3C2**: What was the definition of free draining versus restricted in "Schaffitel et al. (2019) classified the PPs into free-draining PPs"? Please give a one sentence definition

Thank you for pointing out the missing definition. In the revised manuscript, we clarified this point by adding the following:

The classification applied in Schaffitel et al. (2019), is based on a combination of statistical analysis and visual inspection. Plots were classified as "restricted drainage" when soil moisture reached saturation frequently during rain events and remained saturated even after the end of rainfall. In contrast, plots which showed a fast recession of soil moisture were classified as "free drainage". Thereby, the fast soil moisture recession indicates a high hydraulic conductivity of underlying soil layers.

**R3C3:** Do the case study pavements have a porous reservoir layer?

According to the local construction authority, the PPs of the case study should not have a porous reservoir layer. Except two plots, this is in accordance with the observations made during field works. Only at two plots (CP12 and CP2) we encountered coarse gravel underneath the pavement layer which could serve as kind of porous reservoir layer. Those two plots are located on a private parking lot.

We clarified this point in the revised manuscript.

**R3C4:** Were there soil hydraulic conductivity measurements for the case study section done prior to installation of the reservoir layer as a check?

This is an interesting question which we clarified in the revised manuscript. Unfortunately, information is neither available for construction works, nor for preceding measurements. We therefore planned to extract undisturbed soil samples for determining the hydraulic conductivity function from multistep-outflow experiments in the laboratory. Due to the high fraction of soil skeleton and due to the high soil compaction it was impossible to extract undisturbed soil samples. However, for the PPs of our study the soil hydraulic conductivity of underlying soil layers should be high since all PPs were classified as "free drainage" (see R3C").

**R3C5:** Pg 24 line 25, Surface permeability is highly variable across a permeable pavement surface at a scale larger than most surface permeability measuring devices. Generally, not a problem until whole surface clogs because on the same pavement the areas of high permeability areas can handle the flow from low permeability areas nearby. Example: https://www.sciencedirect.com/science/article/pii/S0301479711003525?via%3Dihub Was that also seen in the cited references?

Indeed, there are various studies showing the variability of surface clogging across a permeable pavement surface (e.g. Razzaghmanesh and Beecham, 2018; Sañudo-Fontaneda et al., 2014). Factors controlling the surface clogging of PPs include age, traffic load, maintenance measures, surrounding land use, joint proportion and filling material of joints (Boogaard, Lucke, & Beecham, 2014; Winston et al., 2016). Previous studies showed, that surface clogging occurs mainly in the first years after the construction (e.g. Boogaard et al., 2014b; Borgwardt, 2006; Lucke and Beecham, 2011). The effect of run-on from surrounding surfaces on PP clogging was investigated e.g. by Razzaghmanesh and Borst (2018).

Following the specific comment of reviewer#1, we removed subsection 4.2 from the manuscript. However, to clarify this point, we added the following paragraph to subsection 2.2 of our manuscript.

At the plots of our study, infiltration experiments were performed only once at the beginning of the study period. Due to the lack of successive infiltration experiments, a direct quantification of the clogging progress over the study period is not possible. However, soil moisture time series should allow for an indirect assessment, since surface clogging affects the infiltration capacity which in turn affects soil moisture dynamics. In this way, Razzaghmanesh and Borst (2018), used soil moisture measurements to study clogging dynamics of a PP surface.

For the PPs of our study, we analyzed the measured soil moisture dynamics over the study period. Since we did not observe a change in dynamics over time, we expect that the state of surface clogging remained more or less constant over the study period. One possible explanation therefore might be that none of the PPs was newly build and therefore all plots were already clogged at the beginning of the study period.

**R3C6:** Any recommended next steps for FluSM and potential improvements

Thank you for this comment. Potential improvements include adaptions for the application on sites with vegetation cover and the consideration of horizontal subsurface flow.

We included these points in the revised manuscript according to our answers on R1C1 and R3C7. Concerning recommendations for next steps, we refer to our answer on R3C8.

**R3C7:** Possibility to extend FluSM to account also for horizontal flow, which might be important for estimating possible effects on surrounding infrastructure

Indeed, this is a very interesting point, which we took into account by adding the following paragraph to the manuscript (in conjunction with R3C1):

In case horizontal subsurface flow at the bottom of the PP system is of interest, an extension of FluSM is possible. In a parsimonious approach, the saturated hydraulic conductivity of the underlying soil layer could be used as single parameter to describe the partitioning between deep percolation and horizontal subsurface flow at this border.

**R3C8:** Recommendations to implement the model in practice

Thank you for this remark, which in the revised manuscript is considered by the following paragraph:

The FluSM approach allows deriving continuous water fluxes from soil moisture and meteorological measurements. Compared to direct measurements of soil hydrological fluxes, this poses a relative easy and cheap way for water balance studies and is especially valuable for fields with limited soil hydrologic knowledge (e.g. missing soil hydrologic parameters or lack of knowledge on the correct representation of processes). In this way, we successfully applied FluSM to derive long-term, high resolution hydrological fluxes for 15 different PPs under field conditions. So far, such data were obtained only by costly lysimeter studies. Besides the application for water balance studies, FluSM may also be beneficial for studying soil hydrological processes and contribute to an increased data availability for model validation purposes. In the future, data-driven derivations of soil hydrological fluxes might serve as a simulation benchmark for the application of process based hydrological models. Regarding the ever-increasing availability of soil moisture data on different spatial scales, the demand of such parsimonious approaches should increase.

Comments on presentation
**P2, L23**: change to "enable the calculation"

Acknowledged

**P3, L18:** change to "lead to an improved"

Acknowledged

**P14, L4:** change "fist" to "first"

Acknowledged

**Literature**

Boogaard, F., Lucke, T., & Beecham, S. (2014). Effect of age of permeable pavements on their infiltration function. *Clean - Soil, Air, Water*, *42*(2), 146–152. https://doi.org/10.1002/clen.201300113

Boogaard, F., Lucke, T., van de Giesen, N., & van de Ven, F. (2014). Evaluating the infiltration performance of eight dutch permeable pavements using a new full-scale infiltration testing method. *Water (Switzerland)*, *6*(7), 2070–2083. https://doi.org/10.3390/w6072070

Borgwardt, S. (2006). Long-Term In-Situ Infiltration Performance of Permeable Concrete Block Pavement. *8th International Conference on Concrete Block Paving, November 6-8, 2006 San Francisco, California USAnternational Conference on Concrete Block Paving*, 149–160. http://citeseerx.ist.psu.edu/viewdoc/download?doi=10.1.1.365.9174&rep=rep1&type=pdf

Cristiano, E., Veldhuis, M. C. Ten, & Van De Giesen, N. (2017). Spatial and temporal variability of rainfall and their effects on hydrological response in urban areas - A review. *Hydrology and Earth System Sciences*, *21*(7), 3859–3878. https://doi.org/10.5194/hess-21-3859-2017

FGSV. (2012). *Richtlinien für die Standardisierung des Oberbaus* (Ausgabe 2012).

Hillel, D. (1998). *Environmental Soil Physics*. Acad. Press.

Illgen, M. (2009). Das Versickerungsverhalten durchlässig befestigter Siedlungsflächen und seine urbanhydrologische Quantifizierung. In *Fachbereich Architektur/Raum- und Umweltplanung/Bauingenieurwesen: Vol. PhD*.

Lucke, T., & Beecham, S. (2011). Field investigation of clogging in a permeable pavement system. *Building Research and Information*, *39*(6), 603–615. https://doi.org/10.1080/09613218.2011.602182

Mansell, M., & Rollet, F. (2009). The effect of surface texture on evaporation, infiltration and storage properties of paved surfaces. *Water Science and Technology*, *60*(1), 71–76. https://doi.org/10.2166/wst.2009.323

Niemczynowicz, J. (1999). Urban hydrology and water management – present and future challenges. *Urban Water*, *1*(1), 1–14. https://doi.org/10.1016/S1462-0758(99)00009-6

Razzaghmanesh, M., & Beecham, S. (2018). A review of permeable pavement clogging investigations and recommended maintenance regimes. *Water (Switzerland)*, *10*(3). https://doi.org/10.3390/w10030337

Razzaghmanesh, M., & Borst, M. (2018). Investigation clogging dynamic of permeable pavement systems using embedded sensors. *Journal of Hydrology*, *557*, 887–896. https://doi.org/10.1016/j.jhydrol.2018.01.012

Rodriguez, F., Andrieu, H., & Zech, Y. (2000). Evaluation of a distributed model for urban catchments using a 7-year continuous data series. *Hydrological Processes*, *14*(5), 899–914. https://doi.org/10.1002/(SICI)1099-

1085(20000415)14:5<899::AID-HYP977>3.0.CO;2-R

Sañudo-Fontaneda, L. A., Andrés-Valeri, V. C. A., Rodriguez-Hernandez, J., & Castro-Fresno, D. (2014). Field study of infiltration capacity reduction of porous mixture surfaces. *Water (Switzerland)*, *6*(3), 661–669. https://doi.org/10.3390/w6030661

Schaffitel, A., Schuetz, T., & Weiler, M. (2019). A distributed soil moisture, temperature and infiltrometer dataset for permeable pavements and green spaces. *Earth System Science Data Discussions*, 1–27. https://doi.org/10.5194/essd-2019-97

Shuster, W. D., Bonta, J., Thurston, H., Warnemuende, E., & Smith, D. R. (2005). Impacts of impervious surface on watershed hydrology: A review. *Urban Water Journal*, *2*(4), 263–275. https://doi.org/10.1080/15730620500386529

Winston, R. J., Al-Rubaei, A. M., Blecken, G. T., Viklander, M., & Hunt, W. F. (2016). Maintenance measures for preservation and recovery of permeable pavement surface infiltration rate - The effects of street sweeping, vacuum cleaning, high pressure washing, and milling. *Journal of Environmental Management*, *169*, 132–144. https://doi.org/10.1016/j.jenvman.2015.12.026